# LOL-EVE: Predicting Promoter Variant Effects from Evolutionary Sequences

## Abstract

Genetic studies reveal extensive disease-associated variation across the human genome, predominantly in noncoding regions, such as promoters. Quantifying the impact of these variants on disease risk is crucial to our understanding of the underlying disease mechanisms and advancing personalized medicine. However, current computational methods struggle to capture variant effects, particularly those of insertions and deletions (indels), which can significantly disrupt gene expression. To address this challenge, we present LOL-EVE (Language Of Life for Evolutionary Variant Effects), a conditional autoregressive transformer model trained on 14.6 million diverse mammalian promoter sequences. Leveraging evolutionary information and proximal genetic context, LOL-EVE predicts indel variant effects in human promoter regions. We introduce three new benchmarks for indel variant effect prediction in promoter regions, comprising the identification of putatively causal eQTLs, prioritization of rare variants in the human population, and understanding disruptions of transcription factor binding sites. We find that LOL-EVE achieves state-of-the-art performance on these tasks, demonstrating the potential of region-specific large genomic language models and offering a powerful tool for prioritizing potentially causal non-coding variants in disease studies.

## 1 Introduction

The molecular language of life, DNA, has existed for over 4 billion years, constantly subject to evolutionary pressures. This evolution through natural selection can be seen as a series of countless experiments continuously refining the genomic code to maximize organismal fitness. A long standing challenge of computational biology is to use genomic information to learn a mapping between underlying genomic state and the corresponding organism state, i.e. genotype to phenotype. Utilizing evolutionary sequence information for unsupervised phenotype predictions is valuable as it allows assessment of mutational impacts on organism fitness without requiring a priori knowledge of impact mechanisms or experimental work. While substantial progress has been made in developing computational methods to determine how protein variants affect phenotype (Frazer et al., 2021; Hopf et al., 2017; Orenbuch et al., 2023; Su et al., 2024; Notin et al., 2022; 2023), methods for predicting the effects of variants in the rest of the genome, particularly in non-coding regions, are still in their infancy.

Non-coding regions, which composes 99% of the genome, contain thousands of variants linked to human disease (Maurano et al., 2012). These non-coding variants contribute to many rare and undiagnosed diseases that have eluded diagnosis through protein-coding exome sequencing alone (Marwaha et al., 2022). However, identifying whether these non-coding variants are causal of phenotype changes or merely passenger to, or in linkage disequilibrium, with causal variants remains challenging (Abell et al., 2022).

Current approaches to variant effect prediction in non-coding regions primarily examine single nucleotide variants (SNVs), which have been the primary focus due to the relative ease of their detection in whole-genome sequencing (Mullaney et al., 2010; Jiang et al., 2015). While this approach has yielded valuable insights, there is an opportunity to expand our focus to include insertions and deletions (indels), a vast and important source of genetic variation (Li et al., 2023). Several studies suggest that the probability of individual SNVs having a large effect at an organismal scale is rel-

atively low, especially in non-coding regions (Kircher et al., 2014; Short et al., 2018). This lower impact is partly due to the redundancy built into biological systems and the generally smaller effect sizes of non-coding variants (Zhu et al., 2017). However, there is a considerable amount of heritability in promoter regions, suggesting that these effects may be due to larger variants or the cumulative impact of multiple SNVs (Gazal et al., 2017; Finucane et al., 2015).

Furthermore, many methods have relied on expression or chromatin accessibility data, highly informative in specific biological contexts (Smedley et al., 2016) – however this data is difficult, and sometimes impossible, to gather. As such, developing complementary methods that can make predictions in biological scenarios where such data is unavailable is valuable to the community. Promoter variation likely accounts for a significant percentage of undiscovered causes of disease (Maurano et al., 2012; Albert & Kruglyak, 2015), although research to date has revealed only small effects on clinical outcomes and gene expression (Gamazon et al., 2018; GTEx Consortium et al., 2020). Recent research has shown that the orientation and order of transcription factor (TF) binding sites are major drivers of gene regulatory activity (Georgakopoulos-Soares et al., 2023), necessitating a method that can predict the effects of large insertions or deletions.

We hypothesize that expanding the scope of variant effect prediction to include indels, particularly in promoter regions, could lead to the discovery of variants with larger phenotypic effects (Zheng et al., 2024; Chiang et al., 2017). This approach will potentially identify previously overlooked sources of genetic variation with significant phenotypic impacts, contributing to a deeper understanding of rare and undiagnosed diseases and potentially uncovering new pathways for diagnosis and treatment.

In this paper, we present LOL-EVE (Language Of Life across EVolutionary Effects), a novel genomic large language model designed to address the challenges of predicting indel variant effects in promoter regions. Our key contributions are as follows:

- We develop LOL-EVE, **a 235 million parameter conditional generative model of promoter evolution** for predicting variant effects (§ 3.1);
- We construct a dataset of **14.6 million sequences** comprising almost 20 thousand 1kb promoter region sequences from 447 species across mammalian evolution identified in the Zoonomia project (Christmas et al., 2023) (§ 3.2);
- We create and introduce **three new benchmarks** specifically designed for zero-shot indel variant effect prediction in promoter regions, encompassing rare indel detection, causal variant prioritization and TF binding site disruption (§ 4).

This work not only advances the field of genomic language models but also provides a powerful tool for studying the impact of non-coding variants on gene regulation and disease.

## 2   BACKGROUND

Here, we broadly categorized methods for modeling genomic sequences into alignment-free, alignment-based, and sequence-to-activity. While this paper focuses on unsupervised models for predicting evolutionary sequence fitness, we briefly touch on all three categories.

**Alignment-free methods:** A growing number of unsupervised language models (LMs) for eukaryotic genomic DNA have been proposed, including DNABERT (Ji et al., 2021; Zhou et al., 2024), Nucleotide Transformer (Dalla-Torre et al., 2023), HyenaDNA (Nguyen et al., 2023), and Caduceus (Schiff et al., 2024). While having some differences in their architectures, training objectives, and training data, these models are all fully unsupervised and trained only on genome-wide data (Benegas et al., 2024b). While LMs have shown utility in some downstream prediction tasks, their performance in variant effect prediction varies. Independent benchmarks have revealed that models trained on genome-wide data learn different aspects of the genome to varying extents, sometimes focusing on splice site patterns and other times on regulatory elements, in ways that are difficult to predict (Marin et al., 2024; Li et al., 2024).

An alternative approach involves specialized LMs trained on local genomic regions, such as plant promoters or fungal 5' and 3' regions (Levy et al., 2022; Gankin et al., 2023). These models reliably capture regulatory motifs and learn embeddings useful for downstream tasks. Recently, Vilov & Heinig (2024) proposed and evaluated several 3'UTR-specific language models for the human

genome. Their study showed that these region-specific models often outperformed genome-wide models and even conservation-based approaches like PhyloP on various tasks, including variant effect prediction.

**Alignment-based methods:** Multiple sequence alignments (MSAs) offer a powerful approach to understanding natural sequence variation, enabling the identification of potentially non-neutral mutations with likely functional consequences. PhyloP (Pollard et al., 2010) is an MSA-based statistical method that assigns a conservation score to each position in a sequence and compares observed substitutions to those expected under a neutral evolution model. GPN-MSA (Benegas et al., 2024a), a more recent development, combines whole-genome alignments with a genomic LM approach. Trained to reconstruct masked nucleotides given an MSA as input, GPN-MSA has shown improvement in SNV effect prediction compared to PhyloP. However, a major limitation of alignment-based approaches is their treatment of positions individually, which doesn't naturally generalize to indel variants.

**Sequence-to-activity models & Meta Predictors:** An alternative approach to unsupervised models of sequences involves training supervised regression or classification models on measurements of sequence activity. These models often use data from high-throughput functional genomics experiments that measure various aspects of genomic function, such as expression initiation or epigenetic modifications. Models like Puffin (Dudnyk et al., 2024) and Enformer (Avsec et al., 2021) have demonstrated an understanding of factors contributing to gene expression in different cell types. Notably, Puffin showed correspondence with evolutionary conservation measures like PhyloP, suggesting its ability to capture biologically relevant sequence features that are not cell type specific (Dudnyk et al., 2024), but it has not been tested on variant effect prediction tasks. However, recent studies by Sasse et al. (Sasse et al., 2023) and Huang et al. (Huang et al., 2023) have shown that the performance of models such as DeepSEA (Zhou et al., 2018), Basenji2 (Kelley et al., 2018), Enformer (Avsec et al., 2021) in explaining expression variation between individuals due to cis-regulatory genetic variants remains limited. Another widely used method, CADD (Combined Annotation Dependent Depletion), integrates numerous diverse genomic annotations into a single deleteriousness score using machine learning (Schubach et al., 2024). However, as Grimm et al. (Grimm et al., 2015) demonstrated, comparative evaluations of variant effect predictors like CADD are complicated by circularity issues in their training and testing datasets. These findings underscore the need for further research to overcome these limitations and enhance our understanding of genetic variant effects in humans.

## 3 LOL-EVE

### 3.1 MODEL ARCHITECTURE

To address the challenge of modeling non-aligned promoter sequences across mammalian evolution for indel variant effect prediction, LOL-EVE learns a generative model over full promoter nucleotide sequences. To incorporate evolutionary context, the model conditions its predictions on the promoter's most proximal gene, species, and clade, such as non-primate mammals and primates (Figure 1A-right). This strategy is implemented using a decoder-only transformer architecture, following the CTRL framework (Keskar et al., 2019) (Figure 1B). The conditioning information is provided as prefix tokens, allowing LOL-EVE to generate and score promoter sequences in a context-aware manner. This approach enables the model to capture both broad evolutionary patterns and species-specific variations in regulatory elements. This clade specificity, as shown in (Figure 1A-mid), can be useful for capturing, in this model, mammal vs. primate-specific constraint, which is shown to be crucial for distinguishing disease-associated regulatory variants. Specifically, *primate-constrained elements* are more likely to harbor regulatory variants tied to human-specific traits and diseases, while *mammal-constrained elements* may underlie conserved regulatory processes across a broader evolutionary scope (Kuderna et al., 2023).

We provide the list of all model hyperparameters used in our final architecture in Table A1. Unlike LMs that use k-mer tokenization schemes to achieve length compression (Dalla-Torre et al., 2023; Zhou et al., 2024), LOL-EVE directly tokenizes the promoter sequence $x$ at base pair resolution. This enables the model to accurately handle insertions and deletions without causing tokenization shifts in the remainder of the sequence.

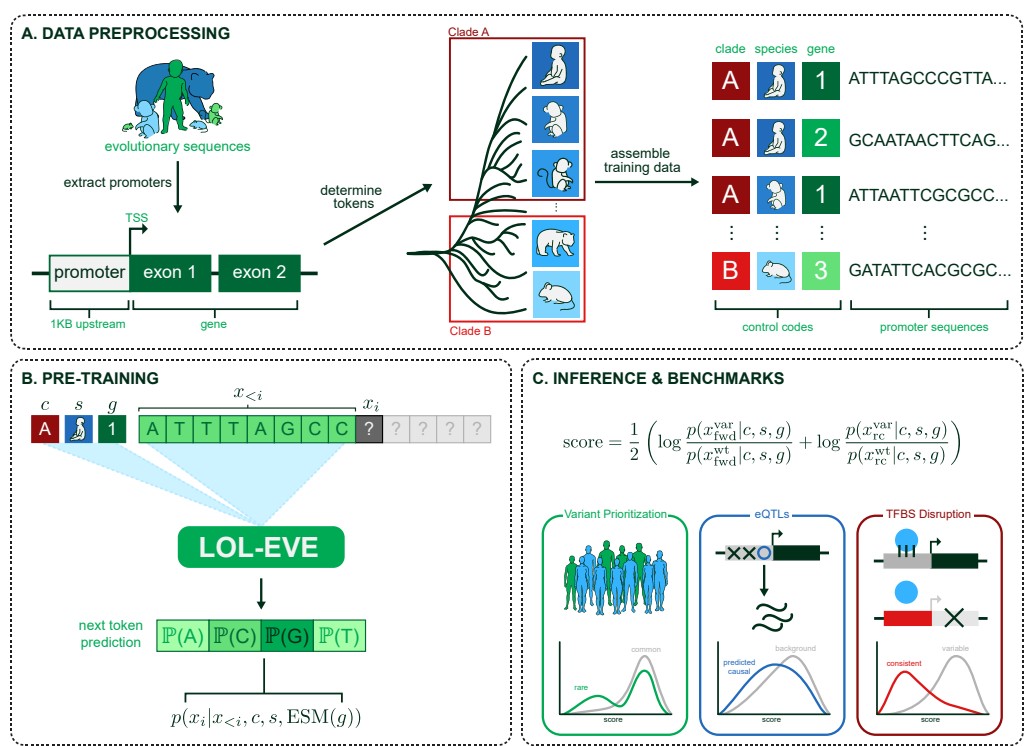

Figure 1: **LOL-EVE approach overview** Figure 1: LOL-EVE approach overview. A. Data pre-processing: Promoter sequences (1KB upstream of first exon) are extracted from evolutionary sequences across mammals. Species are grouped into clades (e.g., Clade A: primates, Clade B: non-primate mammals) and tokenized with control codes for clade, species, and gene identifiers. B. Pre-training: The model performs next-token prediction conditioned on preceding sequence context ($x < i$), control codes for clade (c), species (s), and ESM gene embeddings (g). C. Inference Benchmarks: The model is evaluated on three tasks: (1) Variant Prioritization - distinguishing rare from common variants in human populations, (2) eQTLs - identifying causal expression quantitative trait loci, and (3) TFBS Disruption - predicting transcription factor binding site disruption effects in consistently vs variably expressed genes.

To encode the most proximal gene $g$, we use mean-pooled ESM2 embeddings (`ESM2_t33_650M_UR50D`) (Lin et al., 2023) of a gene's canonical human protein sequence. ESM vectors are kept frozen during training and are projected from dimension 1280 to LOL-EVE's embedding dimension using a learned linear mapping. The ESM-based embedding scheme allows LOL-EVE to generalize to gene tokens unseen during training, which is critical in genomics where chromosome-wise hold outs are typically preferred. The species $s$ and clade $c$ are encoded using learned embeddings. Taken together, LOL-EVE models the autoregressive conditional distribution of a length $L$ promoter as:

$$p(x|c, s, g) = \frac{1}{L} \sum_{i=1}^{L} \log p(x_i|x_{<i}, c, s, \text{ESM}(g)). \tag{1}$$

To prevent overfitting, we apply different data augmentation strategies during training. First, as shown in equation 2 we apply control tag dropout to entice the model to learn representations that are robust to the presence of such tags and mitigate sequence memorization. Second, we augment the training data with reverse complements (rc), enabling LOL-EVE to bidirectionally score promoters as shown in equation 3 and in (Figure 1C).

$$\mathcal{L}(D) = -\sum_{k=1}^{|D|} \log p_\theta(x_i^k|x_{<i}^k, c^k, s^k, g^k, m^k \odot [c^k, s^k, g^k]), \text{ where } m^k \sim \text{Bernoulli}(p) \tag{2}$$

$$\text{score} = \frac{1}{2} \left( \log \frac{p(x_{\text{fwd}}^{\text{var}}|c, s, g)}{p(x_{\text{fwd}}^{\text{wt}}|c, s, g)} + \log \frac{p(x_{\text{rc}}^{\text{var}}|c, s, g)}{p(x_{\text{rc}}^{\text{wt}}|c, s, g)} \right). \tag{3}$$

The score in equation (3) represents the log-likelihood ratio between variant and wildtype sequences. This captures how likely/unlikely the variant sequence is compared to wildtype, given evolutionary patterns learned during pre-training.

## 3.2 TRAINING DATA

Promoters and other regulatory regions generally evolve faster than protein-coding sequences, as regulatory changes can often be more easily tolerated than changes to protein structure and function (Wittkopp & Kalay, 2011). To capture these evolutionarily relevant regulatory signals, particularly those that have evolved recently, we focused on training data from mammals. We curated a promoter dataset across 447 diverse species from the Zoonomia project (Christmas et al., 2023; Kuderna et al., 2023).

Transcription Start Site (TSS) annotations, which are often used to infer promoter regions, are not readily available for most species in our dataset due to several factors. Many of the 447 species lack comprehensive genome annotations, particularly for regulatory regions like promoters. Even in well-annotated species, TSS and promoter definitions can vary significantly across different databases and research groups. To address this, we employed a comparative genomics approach to identify putative promoter regions, leveraging sequence similarity to the first exon of 19,254 protein-coding genes from the NCBI RefSeq human genome annotation (assembly GRCh38.p14, annotation release 109). This strategy allowed us to consistently infer promoter regions across species by aligning known human exonic regions to homologous exons in other species, then extracting sequences upstream of the start of the first exon (which we define as the putative TSS). It's important to note that no genome has "promoter annotations" as such; rather, we use these inferred TSS positions and their upstream sequences as proxies for promoter regions. Importantly, in the human annotations we utilized, the 5'UTR often overlaps with the annotations for exon 1, which influences our definition of putative promoter regions across species. A visual representation of the sequence regions is shown in Figure 1A-left.

Using the HAL toolkit (Hickey et al., 2013), we performed a liftover of these exon coordinates to each species in the Zoonomia project. For each species, exons were retained if their length was at least 50% of the length of the corresponding human exon. This threshold ensured that conserved regions were captured while excluding regions where the alignment is unreliable.

To define promoter regions, we extracted the 1,000 base pairs upstream of each exon start, accounting for the strand orientation of the gene. This conservative approach minimized the risk of including non-promoter sequences but may exclude more distal regulatory elements, a potential caveat of the 1,000 bp window approach. Additionally, in cases where promoter regions from neighboring genes were within 100 base pairs of each other, we merged the coordinates. This merging process ensured that promoter regions were not artificially fragmented due to closely spaced genes.

To gain further insight into the validity of the upstream 1,000 bp approach, we scored all extracted sequences using the Sei promoter score (Chen et al., 2022), which is trained on functional genomics data from humans. Despite Sei being human-based, we found that the promoter scores generalize well across species, showing strong conservation of regulatory elements in many mammalian species. Notably, promoters from species closely related to humans, such as other primates, tend to have higher Sei scores, indicating similar promoter activity, while more distant species still retain significant functional signal, suggesting that core regulatory sequences are preserved across mammals (A1). Further we assessed how the Sei score distributions for 3 groups: Human CDS regions, Human promoters, and our training data compare in (A2). Our training data promoter distribution aligns more closely with the raw Human promoters than the Human CDS regions.

Including reverse complements, this resulted in a dataset of 14.6 million sequences. We employed a chromosome-wise split for development, with chromosome 19 used for validation. Promoters from non-human species were assigned to the respective set based on the chromosome of the human gene used for liftover, thereby ensuring that all instances of a gene are placed in the same partition and no gene information leakage between the training and validation set.

## 4    BENCHMARKS

In the following section, we introduce a collection of benchmark tasks designed to evaluate the unsupervised understanding of promoter variation. There are currently no established benchmarks focused specifically on promoter indel variant effect prediction, making this work a significant contribution to the field. All tasks are evaluated in a zero-shot setting, meaning the models have not been explicitly trained on the specific benchmarks, highlighting their generalization ability to unseen promoter sequences and variant effects. To ensure rigorous and fair model comparisons across these benchmarks, we maintain strict methodological consistency. This includes using standardized scoring approaches across all models (detailed in A.3), implementing identical preprocessing and evaluation pipelines, and focusing exclusively on zero-shot performance without any task-specific training or fine-tuning.

### 4.1    FREQUENCY-BASED INDEL PRIORITIZATION

**Rationale**    Variants that are rare in the human population are generally more likely to be deleterious than common variants, which are more likely to be of neutral consequence (Lohmueller et al., 2008). Rare variants tend to be under stronger selective pressure, and as a result, they are often associated with more severe functional consequences. Therefore, rare variants should be assigned lower variant effect scores, reflecting their low likelihood of observation and possible deleteriousness.

**Task**    Given a collection of indel variants labeled as rare and common based on population-wide incidence, models should assign lower variant effect scores to rare variants. Using predicted model scores, we evaluate the enrichment of low-frequency variants in the top 1% score percentile as the odds ratio.

**Data**    We collect promoter indel variants from gnomAD (Chen et al., 2024) release V4.0, categorizing variants into low-frequency variants and common variants using a mean allele frequency (MAF) threshold of 0.05 (Consortium, 2015), yielding 578,495 low frequency indel variants and 15,137 common indel variants.

### 4.2    PUTATIVELY CAUSAL eQTL PRIORITIZATION

**Rationale**    An expression Quantitative Trait Locus (eQTL) is a genetic variant that is statistically associated with a phenotypic change in gene expression. For some eQTLs, the effect of the variant

on the expression change is putatively causal. Causality can be inferred using fine-mapping approaches, such as SUSIE (Wang et al., 2020b), which yield a posterior inclusion probability (PIP) that quantifies the likelihood of causality. EQTLs can be anywhere in the genome regardless of the position of their affected gene (eGene), eQTLs that are proximal to their eGene are referred to as *cis*.

**Task** As the evolutionary directionality of a variant affecting expression is unclear (a causal change in gene expression may be benign or deleterious), scores are evaluated as |score| in this task. Given a collection of putatively causal and non-causal cis-eQTL indels in promoter regions, models are expected to assign larger effect scores to putatively causal eQTLs because putatively causal variants are more likely to induce meaningful changes in gene regulation compared to non-causal variants. We analyze the difference in score between the two groups as the area under the precision-recall curve (AUPRC) and as the effect size, as quantified by Cohen's d. To make AUPRC values more intuitive, we normalize the AUPRC by dividing it by the baseline AUPRC, which is equivalent to the proportion of putatively causal variants in the dataset as defined in Table A4.

**Data** Putatively Causal eQTLs, fine-mapped credible sets based on SuSiE analysis (Wang et al., 2020a) from 42 individual studies, were aquired from the eQTL Catalogue (Kerimov et al., 2021) and filtered for those falling into our promoter regions defined previously. We subsetted to indel eQTLs in promoter regions, filtering for cis-eQTLs where the eGene is the promoter's proximal gene. We bin the data into putatively causal and background eQTLs using a PIP cutoff of 0.95, yielding 132 putatively causal and 3,949 non-causal variants.

## 4.3 TFBS Disruption

**Rationale** Transcription factors (TFs) are essential regulators of gene expression, binding to specific DNA sequences in promoter regions to control transcriptional activity. Disruptions to TF binding sites (TFBS) can significantly impact gene regulation, particularly in genes with consistent expression across multiple tissues. Genes with consistent expression across tissues are often more intolerant to mutations, suggesting that disrupting TFBS in these genes could have more severe consequences than in genes with variable expression across tissues (Wolf et al., 2023).

**Task** We divide genes into two groups (consistent and variable expression across tissues). For each TF, we score *in silico* variants that completely delete TFBS in both groups. We expect variants disrupting TFBS in consistently expressed genes to be more deleterious than those in variably expressed genes. We assume that deleting the TFBS would be deleterious in this context as the gene of interest would no longer be expressed For each TF, we evaluate whether variants in consistently expressed genes receive lower (more deleterious) scores than variants in variably expressed genes. We report the delta accuracy (observed accuracy minus random accuracy of 0.5) across all TFs.

**Data** The gene groups were constructed using GTEx (Consortium, 2020) data to calculate the coefficient of variation (CV) for gene expression across tissues. The 500 genes with the lowest CV formed the "consistent expression" group, while the 500 genes with the highest CV constituted the "variable expression" group. To identify relevant *in silico* TFBS disruptions, we employed a two-step process:

1. **TF Selection:** We sourced human TFs from the JASPAR CORE (Fornes et al., 2020) database, applying a filter to include only those expressed above 1 TPM in at least 30 tissue types.

2. **TFBS Identification:** Using position-specific scoring matrices (PSSMs) from JASPAR, we scanned promoter sequences for TFBS with scores exceeding 0.8. A TFBS was considered knocked out if, following the deletion of the entire TFBS, the PSSM score in the mutated region fell below 0.8.

In total, we analyzed 340 TFs that met our filtering criteria, resulting in 38,854 deletions for the consistently expressed gene group and 3,790 deletions for the variably expressed gene group.

## 5 RESULTS

We benchmark LOL-EVE against DNA LMs that are applicable to the human genome: HyenaDNA (Nguyen et al., 2023), DNABERT-2 (Zhou et al., 2024), Nucleotide Transformer (NT) (Dalla-Torre et al., 2023), and Caduceus (Schiff et al., 2024). For LMs that make multiple checkpoints available, we focus our discussion on the best performing checkpoint in each experiment, with remaining checkpoints evaluated in section A.4. For scoring, we use the likelihood of autoregressive LMs, and the pseudolikelihood for masked LMs (A.3). For benchmarking PhyloP, we use the score at the position of the indel in the reference genome. Additionally, PhyloP is the only score where low indicates a less conserved region or a region more tolerant to mutation, thus, within this work, we always invert PhyloP scores to maintain consistent directionality for all methods.

### 5.1 FREQUENCY-BASED INDEL PRIORITIZATION

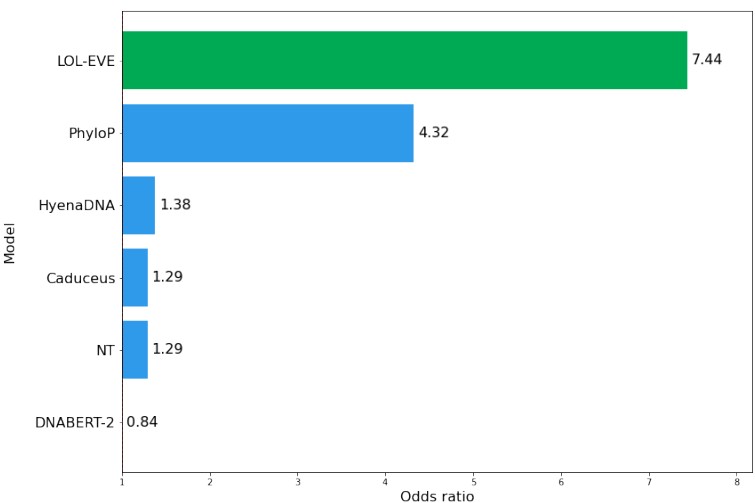

Figure 2: Comparison of odds ratios for low frequency range (0.0 - 0.05) vs. common range($\geq 0.05$) variants across different models at the top 1% score percentile. Ablations are in Figure A3. Best checkpoints: `m-450k`, `ps-131k`, `500m-human-ref`.

LOL-EVE demonstrates superior performance in distinguishing between low-frequency and common indels across various minor allele frequency (MAF) thresholds (Figure A3). As shown in Figure 2, LOL-EVE achieves higher odds ratios compared to other models, particularly for rare variants (MAF $< 0.001$). This suggests that LOL-EVE is more effective at identifying potentially deleterious rare variants in promoter regions. PhyloP also shows good performance, while DNABERT-2, Nucleotide Transformer, Caduceus, HyenaDNA have lower discriminative power for this task.

## 5.2 Causal eQTL Prioritization

Table 1: Performance on the eQTL causal variant prediction task. Effect sizes were calculated as Cohen's d. Complete results are in Table A5. Ablation of metrics vs PIP thresholds Figure A9. Best checkpoints: `ph-131k,s-32k, 2.5B-1000G`

| Model | Effect size (↑) | AUPRC (↑) | Norm. AUPRC (↑) |
|---|---|---|---|
| LOL-EVE | **0.26** | **0.19** | **1.42** |
| Caduceus | 0.21 | 0.17 | 1.28 |
| HyenaDNA | 0.13 | 0.15 | 1.17 |
| NT | 0.11 | 0.15 | 1.11 |
| PhyloP | 0.126 | 0.151 | 1.140 |
| DNABERT-2 | -0.1 | 0.12 | 0.94 |

Table 1 illustrates LOL-EVE's superior performance in distinguishing between causal and background eQTL variants. With the highest effect size (Cohen's d = 0.28) and normalized AUPRC (1.46), LOL-EVE outperforms other models in identifying causal variants. Nucleotide Transformer shows the second-best performance, while DNABERT-2 shows a negative effect size.

## 5.3 TFBS Disruption

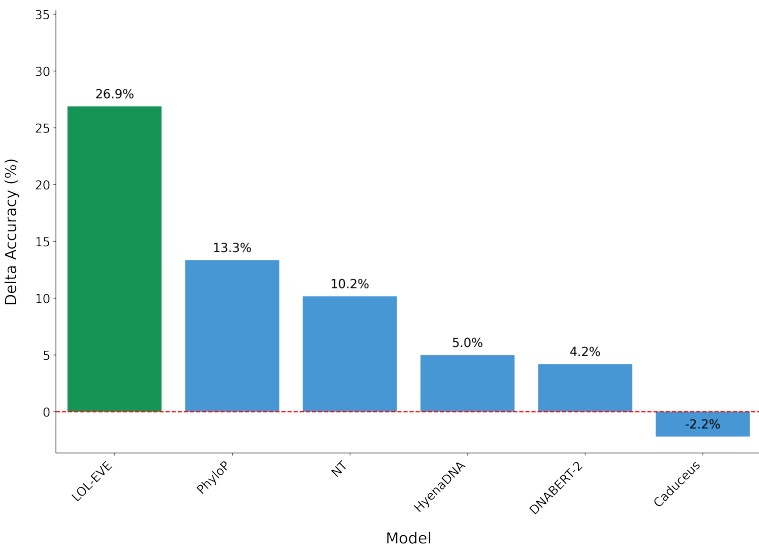

Figure 3: Comparison of delta accuracy scores models in predicting *in silico* TFBS disruptions. All models in Figure A12. Best checkpoints: `small-32k,ps-131k, 2.5B-MS`.

Figure 3 demonstrates LOL-EVE's effectiveness in predicting the impact of *in silico* TFBS disruptions between consistently and variably expressed genes. LOL-EVE outperforms other models in differentiating the potential effects of disruptions on these two gene sets. The model achieves a higher percentage of transcription factors for which it correctly predicts lower disruption scores for consistently expressed genes compared to variably expressed genes. LOL-EVE's ability to capture this biological expectation - that variably expressed genes should be less sensitive to TFBS disruptions than consistently expressed genes - indicates that LOL-EVE can more accurately predict

the functional impact of variations in these regions, aligning well with our understanding of gene regulation dynamics across tissues.

# 6 DISCUSSION & CONCLUSION

LOL-EVE's consistent superior performance across multiple benchmarks demonstrates its potential as a powerful tool for predicting the effects of indel variants in promoter regions. By leveraging evolutionary information and proximal genetic context, LOL-EVE captures important aspects of regulatory genomics that other models may overlook. The model's ability to distinguish between low-frequency and common indels suggests it has learned to identify potentially deleterious variants under negative selection. This capability is particularly valuable for identifying rare variants that may contribute to disease risk, addressing a significant challenge in genomics research. Furthermore, LOL-EVE's strong performance in prioritizing causal eQTLs indicates its potential utility in fine-mapping studies and in elucidating the genetic basis of gene expression variation. LOL-EVE's effectiveness in predicting transcription factor binding site disruptions in consistently or variably expressed genes is especially noteworthy. This suggests that the model has learned to recognize complex patterns in regulatory sequences and can predict the functional impact of variations in these regions with respect to gene expression dynamics. Such capability is likely invaluable in understanding the mechanisms by which non-coding variants contribute to phenotypic variation and disease risk.

The superior performance of LOL-EVE over other models can be attributed to several factors that address limitations in existing approaches. Many current models employ tokenization strategies that may be disrupted by indel changes leading to poor performance in variant effect prediction tasks. LOL-EVE's base-pair-level tokenization allows it to handle indels more effectively, maintaining sequence integrity even in the presence of insertions or deletions. Additionally, models trained solely on the human genome or on full genomes may lack the specific context necessary for accurate promoter region analysis. LOL-EVE's focus on promoter regions across multiple species provides it with a rich evolutionary context, allowing it to capture subtle regulatory patterns that may be missed by more generalized models. This specialized training approach enables LOL-EVE to better understand the functional importance of specific sequence motifs in promoter regions. Furthermore, models relying on alignments for training or inference may struggle with promoter regions, which often do not align perfectly across species due to their rapid evolution. LOL-EVE's alignment-free approach circumvents this issue, allowing it to capture regulatory information without being constrained by alignment artifacts. This is particularly important for analyzing rapidly evolving regulatory regions where traditional alignment-based methods may fail to capture important functional relationships.

While LOL-EVE shows promising results, there is still significant room for improvement. Future work could focus on incorporating additional sources of biological information, such as more extensive genomic sequencing across mammalian evolution, to further enhance the model's predictive power. Moreover, experimental validation of LOL-EVE's predictions will be crucial in establishing its reliability for use in clinical and research settings. By addressing these limitations of existing models, LOL-EVE represents a significant step forward in our ability to predict and understand the effects of genetic variations in promoter regions. Its performance across diverse benchmarks reflecting critical challenges in disease genetics suggests that this approach of combining evolutionary information with specialized training on promoter regions could set a new standard for genomic language models in regulatory genomics.

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

# A    APPENDIX

## A.1    MODEL DETAILS

Table A1: The hyperparameters of the LOL-EVE model.

| Hyperparameter | Value |
|---|---|
| Dimension | 768 |
| Layers | 12 |
| Heads | 12 |
| Feedforward dimension | 8192 |
| Learning rate | $1e^{-5}$ |
| Batch size | 16 |
| Epochs | 7 |

## A.2    TRAINING DATA

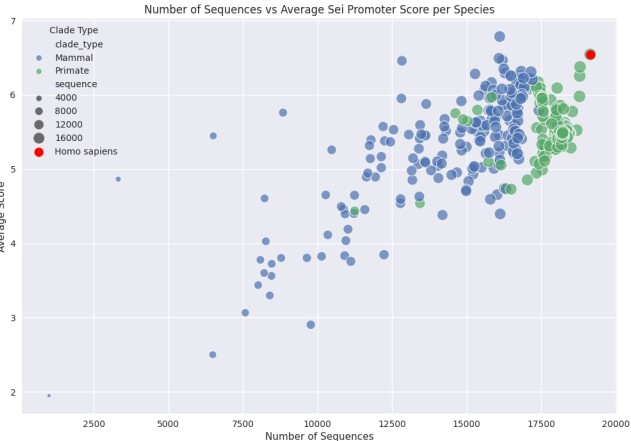

Figure A1: Average Promoter Sei scores plotted against the number of promoter sequences gathered for model training from the comparative genomics analysis conducted with the HAL suite. Clade types are specified by color and the red dot represents Homo sapiens. The maximum number of sequences per species is 19,254. Point sizes reflect the number of sequences.

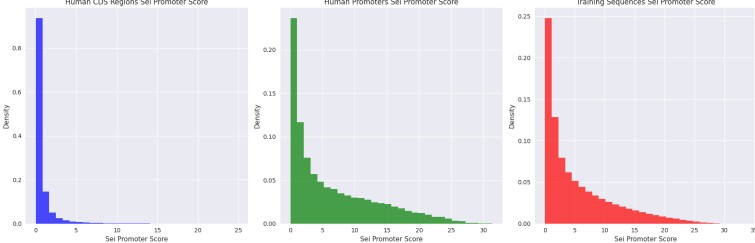

Figure A2: Average Promoter Sei scores were plotted for Human CDS regions, Human promoter regions, and all of the promoter data used gathered for training.

## A.3 Baseline details

### A.3.1 Autoregressive models

Autoregressive LMs assign scores to sequences $s$ using their log likelihood

$$p(s) = \frac{1}{n} \sum_{i=1}^{n} \log p(s_i|s_{<i}). \tag{4}$$

**HyenaDNA**  HyenaDNA uses base pair tokenization. For computing the cross entropy, we subset the logits and labels to the dimensions of actual nucleotides $x \in \{A, G, C, T, N\}$ and exclude special tokens. We ignore the final EOS position when taking the mean over the sequence.

### A.3.2 Masked language models

For computational efficiency, we evaluate bidirectional masked LMs using their pseudo log likelihood,

$$p(s) = \frac{1}{n} \sum_{i=1}^{n} \log p(s_i|s). \tag{5}$$

**Caduceus**  Caduceus uses base pair tokenization. For computing the cross entropy, we subset the logits and labels to the dimensions of actual nucleotides $x \in \{A, G, C, T, N\}$ and exclude special tokens. We do not apply any masking.

**Nucleotide Transformer**  Nucleotide Transformer uses 6-mer tokenization. For computing the cross entropy, we subset the logits and labels to the dimensions of the 6-mer and five trailing single-base tokens and exclude special tokens. We do not apply any masking.

**DNABERT-2**  DNABERT-2 uses byte pair tokenization. For computing the cross entropy, we subset the logits and labels to the dimensions of the BPE tokens and the [UNK] token which represents $N$. Remaining special tokens are excluded. We do not apply any masking.

### A.3.3 Alignment-based approaches

**PhyloP**  As they are based on an MSA, PhyloP scores are not naturally amenable to indel variants, as a change in sequence length by insertion or deletion cannot be modeled by column-wise scores. We follow gnomAD's approach to computing PhyloP scores: For any indel, the PhyloP score of the position in the reference genome at which the indel occurs is used for the indel as a whole. Note that this inherently does not consider the actual sequence consequence of the indel - it only reflects the conservation of the position at which the indel occurs.

## A.4 Extended Results on Benchmark datasets

### A.4.1 Frequency-Based Indel Prioritization

Table A2: MAF cutoffs and corresponding counts.

| Low-freq range | Common range | Low-freq count | Common count |
|---|---|---|---|
| 0.0 - 0.05 | $\geq 0.05$ | 578,495 | 15,137 |
| 0.0 - 0.01 | $\geq 0.01$ | 563,533 | 30,099 |
| 0.01 - 0.05 | $\geq 0.05$ | 48,972 | 15,137 |
| 0.001 - 0.01 | $\geq 0.01$ | 34,010 | 30,099 |

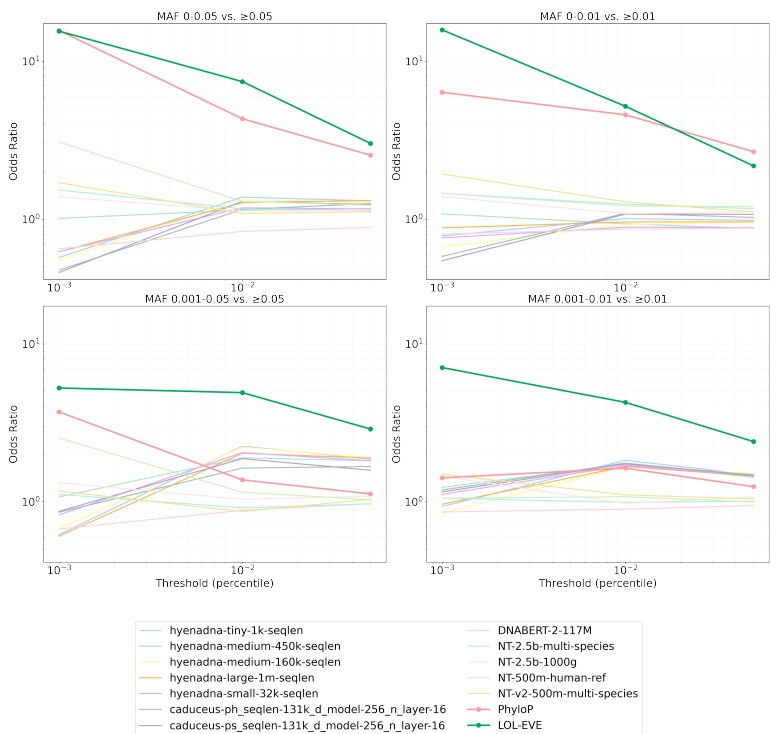

Figure A3: Ablation of odds ratios values for models across a variety of MAF cutoffs for low frequency and common variants.

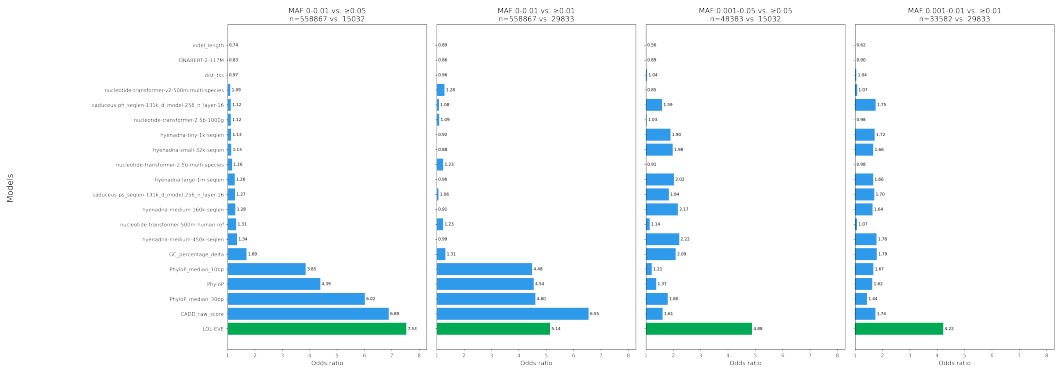

Figure A4: Comparison of odds ratios for rare indel identification across different minor allele frequency (MAF) thresholds. Each panel shows results for different MAF cutoff comparisons, with sample sizes indicated (n=rare vs common variants). Tools are compared based on their ability to distinguish between rare and common variants, measured as odds ratios at top 1% score percentile.

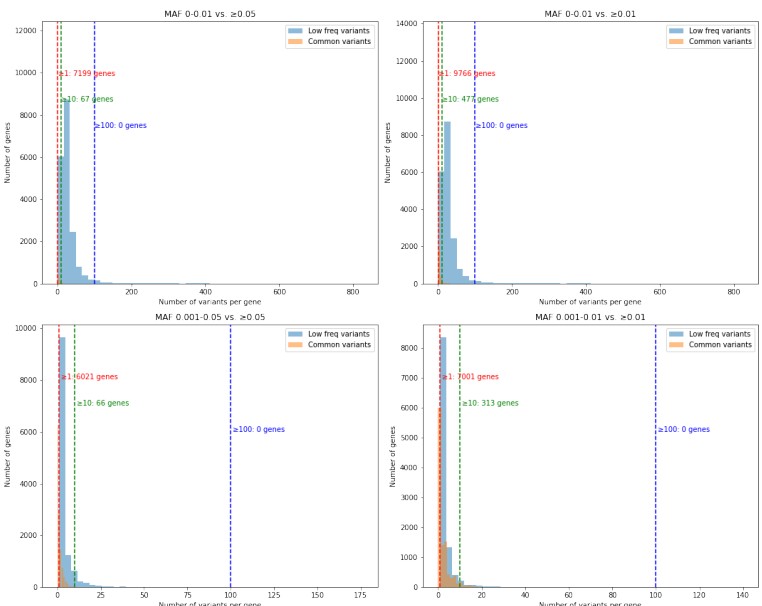

Figure A5: Distribution of variants per gene at different Minor Allele Frequency (MAF) thresholds, comparing low frequency (blue) and common variants (orange). Vertical lines mark genes with 1 (red), 10 (green), and 100 (blue) variants for both classes. Most genes contain few variants, with counts decreasing exponentially.

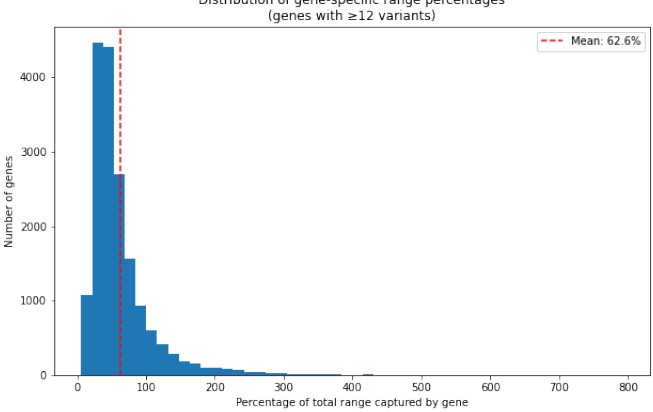

Figure A6: Distribution of gene-specific LOL-EVE score ranges (max-min) normalized to total range (2.5th-97.5th percentiles). Analysis includes genes with 12 variants(10th percentile threshold). Mean range: 62.6% (red dashed line).

Table A3: Performance of model checkpoints on eQTL causal variant prediction, grouped by model type and ranked by NAUPRC.

| Model | C.d | AUPRC | NAUPRC | C.d>5bp | AUPRC>5bp | NAUPRC>5bp |
|---|---|---|---|---|---|---|
| *DNA Language Models* | | | | | | |
| **LOL-EVE** | **0.261** | **0.187** | **1.416** | **0.375** | **0.355** | **1.480** |
| Hyena-32k | 0.134 | 0.154 | 1.166 | 0.103 | 0.261 | 1.085 |
| Hyena-1m | 0.117 | 0.150 | 1.139 | 0.105 | 0.264 | 1.099 |
| Hyena-1k | 0.116 | 0.149 | 1.131 | 0.067 | 0.253 | 1.052 |
| Hyena-450k | 0.114 | 0.148 | 1.122 | 0.105 | 0.259 | 1.077 |
| Hyena-160k | 0.118 | 0.148 | 1.120 | 0.103 | 0.256 | 1.064 |
| NT-2.5B | 0.079 | 0.147 | 1.113 | 0.243 | 0.298 | 1.242 |
| NT-500M | 0.112 | 0.147 | 1.111 | 0.151 | 0.271 | 1.130 |
| NT-500M-H | -0.047 | 0.128 | 0.966 | -0.070 | 0.230 | 0.958 |
| DNABERT-2 | -0.106 | 0.123 | 0.935 | -0.114 | 0.233 | 0.971 |
| NT-2.5B-MS | -0.098 | 0.122 | 0.928 | -0.245 | 0.205 | 0.851 |
| Caduceus-ph | 0.205 | 0.169 | 1.282 | 0.225 | 0.299 | 1.246 |
| Caduceus-ps | 0.189 | 0.165 | 1.253 | 0.150 | 0.280 | 1.164 |
| *Non gLM* | | | | | | |
| GC% change | 0.178 | 0.175 | 1.323 | 0.122 | 0.262 | 1.090 |
| dist_TSS | 0.038 | 0.154 | 1.164 | 0.050 | 0.278 | 1.156 |
| FATHMM-indel | 0.105 | 0.151 | 1.142 | -0.208 | 0.234 | 0.974 |
| PhyloP | 0.126 | 0.151 | 1.140 | 0.159 | 0.272 | 1.134 |
| PhyloP_median_10bp | -0.014 | 0.135 | 1.020 | 0.248 | 0.310 | 1.290 |
| PhyloP_median_30bp | -0.006 | 0.134 | 1.011 | 0.220 | 0.294 | 1.226 |
| *Sequence-to-Expression Models* | | | | | | |
| Enformer-single-pos-entropy | 0.062 | 0.142 | 1.079 | 0.114 | 0.255 | 1.060 |
| Enformer-avg-entropy | 0.047 | 0.142 | 1.078 | 0.050 | 0.239 | 0.997 |

### A.4.2 CAUSAL EQTL PRIORITIZATION

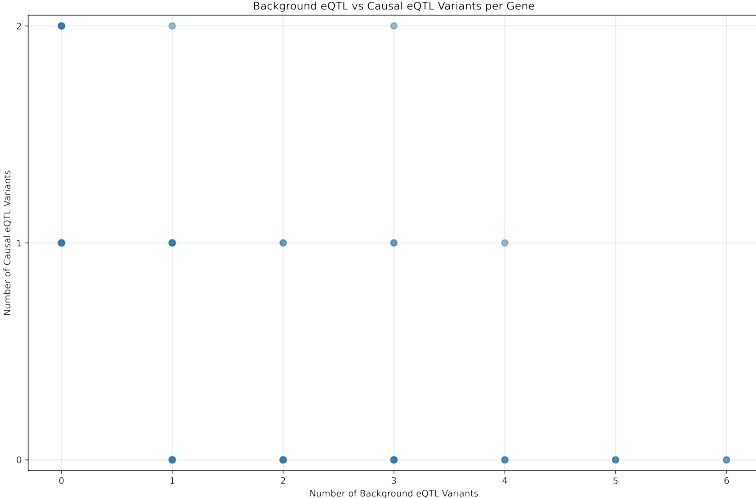

Figure A7: Distribution of causal versus background eQTL variants across genes, showing most genes contain 0-2 causal variants regardless of their background variant count (0-6 variants)

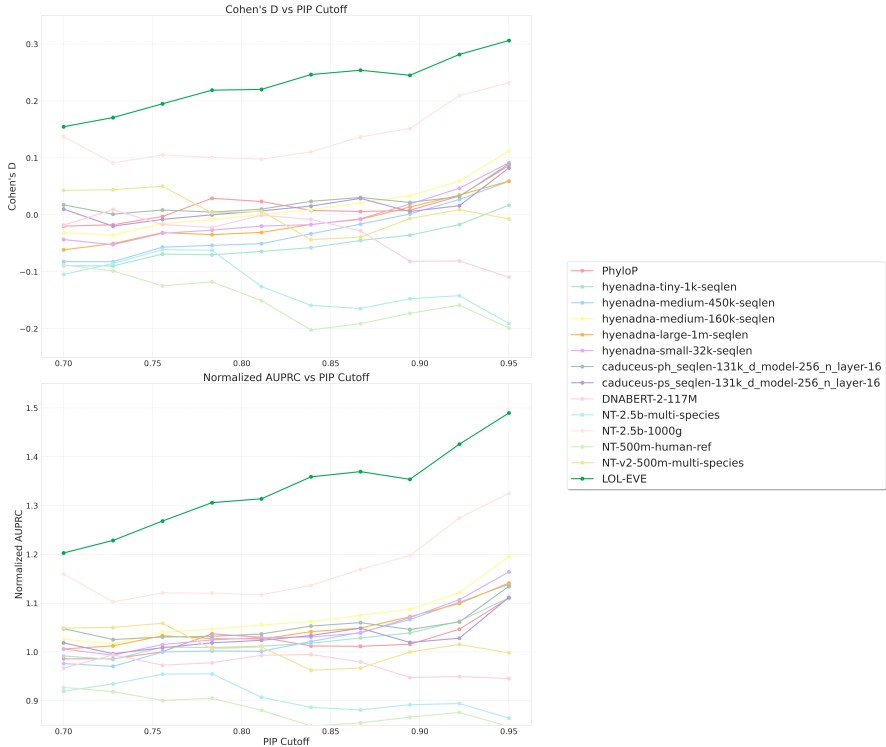

Figure A9: An ablation of PIP scores ranging from .7 to .95 with a step size of 0.025 for the Causal eQTL Prioritization Benchmark metrics Cohen's D and Normalized AUPRC.

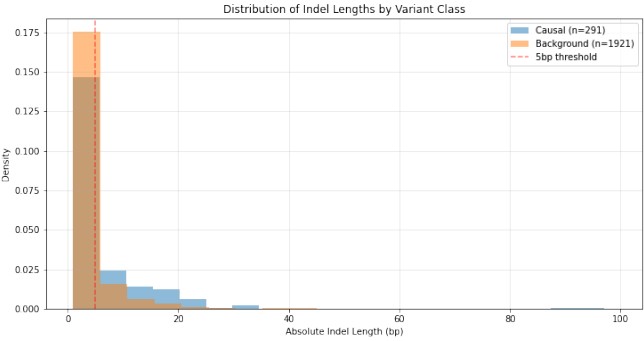

Figure A8: Distribution of insertion/deletion (indel) lengths in base pairs (bp) for causal (n=291) and background (n=1921) eQTL variants. The histogram shows the density of absolute indel lengths, with causal variants shown in blue and background variants in orange. A 5bp threshold is marked by the red dashed line

Table A4: Posterior inclusion probability (PIP) cutoff and corresponding causal and background counts.

| PIP cutoff | Causal count | Background count |
|---|---|---|
| 0.95 | 132 | 3949 |
| 0.92 | 151 | 3930 |
| 0.89 | 162 | 3919 |
| 0.87 | 173 | 3908 |
| 0.84 | 181 | 3900 |
| 0.81 | 188 | 3893 |
| 0.78 | 194 | 3887 |
| 0.76 | 205 | 3876 |
| 0.73 | 215 | 3866 |
| 0.70 | 225 | 3856 |

### A.4.3  TFBS DISRUPTION

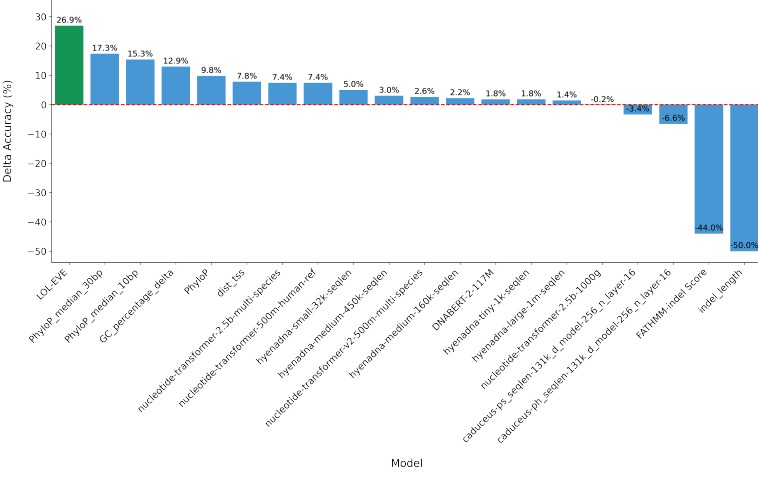

Figure A10: All model scores shown for TFBS disruption Benchmark.

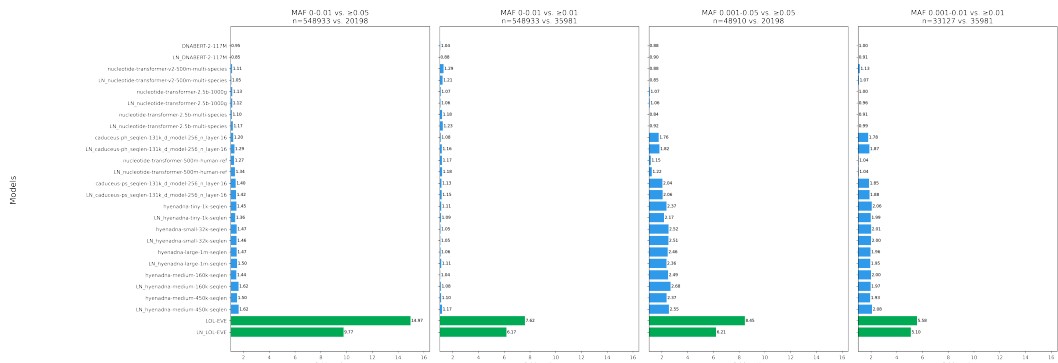

Figure A11: Comparison of odds ratios for rare indel identification across different minor allele frequency (MAF) thresholds. Each panel shows results for different MAF cutoff comparisons, with sample sizes indicated (n=rare vs common variants). Tools are compared based on their ability to distinguish between rare and common variants, measured as odds ratios at top 1% score percentile. Here LN_ indicates that the models scores were length normalized.

Table A5: Performance of model checkpoints on the eQTL causal variant prediction task.

| Model | C.d | AUPRC | NAUPRC | C.d>5bp | AUPRC>5bp | NAUPRC>5bp |
|---|---|---|---|---|---|---|
| **LOL-EVE_LN** | **0.277** | **0.192** | **1.460** | **0.353** | **0.350** | **1.455** |
| LOL-EVE | 0.266 | 0.190 | 1.447 | 0.307 | 0.334 | 1.388 |
| Caduceus-ph_LN | 0.206 | 0.169 | 1.283 | 0.225 | 0.299 | 1.246 |
| Caduceus-ph | 0.188 | 0.164 | 1.247 | 0.188 | 0.289 | 1.203 |
| Caduceus-ps_LN | 0.188 | 0.165 | 1.251 | 0.150 | 0.280 | 1.164 |
| Caduceus-ps | 0.174 | 0.160 | 1.218 | 0.119 | 0.268 | 1.117 |
| Hyena-32k_LN | 0.135 | 0.153 | 1.167 | 0.103 | 0.261 | 1.085 |
| Hyena-32k | 0.114 | 0.146 | 1.113 | 0.061 | 0.240 | 1.000 |
| Hyena-1m_LN | 0.118 | 0.150 | 1.139 | 0.105 | 0.264 | 1.099 |
| Hyena-1m | 0.101 | 0.144 | 1.096 | 0.068 | 0.244 | 1.016 |
| Hyena-1k_LN | 0.117 | 0.149 | 1.132 | 0.067 | 0.253 | 1.052 |
| Hyena-1k | 0.095 | 0.144 | 1.097 | 0.020 | 0.239 | 0.993 |
| Hyena-450k_LN | 0.114 | 0.148 | 1.122 | 0.105 | 0.259 | 1.077 |
| Hyena-450k | 0.094 | 0.145 | 1.098 | 0.066 | 0.249 | 1.037 |
| Hyena-160k_LN | 0.118 | 0.147 | 1.120 | 0.103 | 0.256 | 1.064 |
| Hyena-160k | 0.099 | 0.145 | 1.099 | 0.064 | 0.248 | 1.034 |
| NT-2.5B_LN | 0.082 | 0.147 | 1.116 | 0.243 | 0.298 | 1.242 |
| NT-2.5B | 0.093 | 0.144 | 1.093 | 0.135 | 0.267 | 1.113 |
| NT-500M_LN | 0.108 | 0.146 | 1.107 | 0.151 | 0.271 | 1.130 |
| NT-500M | 0.063 | 0.142 | 1.077 | 0.211 | 0.300 | 1.250 |
| NT-500M-H_LN | -0.045 | 0.127 | 0.969 | -0.070 | 0.230 | 0.958 |
| NT-500M-H | -0.051 | 0.127 | 0.965 | -0.082 | 0.229 | 0.953 |
| DNABERT-2_LN | -0.102 | 0.123 | 0.936 | -0.114 | 0.233 | 0.971 |
| DNABERT-2 | -0.091 | 0.124 | 0.940 | -0.078 | 0.238 | 0.990 |
| NT-2.5B-MS_LN | -0.091 | 0.123 | 0.932 | -0.245 | 0.205 | 0.851 |
| NT-2.5B-MS | -0.087 | 0.122 | 0.930 | -0.238 | 0.205 | 0.852 |

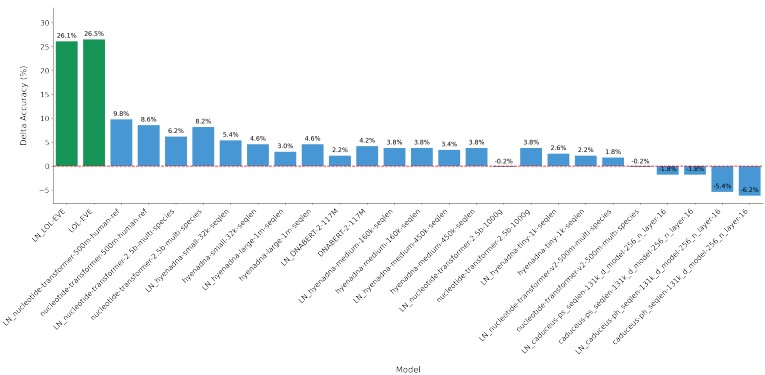

Figure A12: All model scores shown for TFBS disruption Benchmark comparing Length normalization vs nonnormalized scores.

### A.4.5 SNV EXPANSION EXPERIMENT

Table A6: Performance comparison of models on prediction task, grouped by type and ranked by NAUPRC.

| Type | Model | Cohen's d | AUPRC | NAUPRC |
|---|---|---|---|---|
| Non gLM | **PhyloP** | **0.205** | **0.133** | **1.263** |
| | dist_tss | 0.031 | 0.129 | 1.227 |
| Nucleotide | NT-2.5B | 0.066 | 0.113 | 1.075 |
| | NT-1.5B | 0.051 | 0.112 | 1.061 |
| | NT-500M | 0.019 | 0.108 | 1.026 |
| | NT-500M-H | -0.017 | 0.104 | 0.990 |
| HyenaDNA | hyenadna-tiny-1k | -0.135 | 0.098 | 0.926 |
| | hyenadna-medium-450k | -0.081 | 0.101 | 0.964 |
| | hyenadna-medium-160k | -0.104 | 0.100 | 0.948 |
| | hyenadna-large-1m | -0.086 | 0.101 | 0.961 |
| | hyenadna-small-32k | -0.113 | 0.099 | 0.939 |
| Caduceus | caduceus-ph | -0.090 | 0.100 | 0.948 |
| | caduceus-ps | -0.078 | 0.100 | 0.954 |
| | DNABERT-2 | 0.075 | 0.112 | 1.063 |
| | LOL-EVE | 0.054 | 0.112 | 1.059 |

