# OpenReview forum: "LOL-EVE: Predicting Promoter Variant  Effects from Evolutionary Sequences"
_ICLR.cc/2025/Conference — ICLR 2025 Conference Withdrawn Submission_

### Official Review · Reviewer_2Zmu · 2024-11-02

**Soundness:** 4
**Presentation:** 4
**Contribution:** 3
**Rating:** 5
**Confidence:** 4

**Summary:**

This paper presents a conditional autoregressive transformer model trained on a large dataset of mammalian promoter sequences, which can be used to predict the effects of genetic variants in promoter sequences. The authors benchmark the model on three indel (insertion/deletion) variant effect prediction tasks, in contrast to the single nucleotide variant tasks that models are more commonly benchmarked on. They outperform a non-deep learning conservation baseline as well as previous genomic deep learning models trained genome-wide rather than on only promoter sequences.

**Strengths:**

- The paper is very clear, well written and well motivated.
- The paper is highly original and significant. While a number of different DNA language models - which are cited by the paper - have been proposed, they generally take the approach of training on genome-wide sequences. Therefore, the approach presented here to train a focused model on promoter sequences, as well as conditioning on clade, species, and gene, is quite novel.
- The focus on indel prediction tasks is also novel, and an important understudied area, as most prior studies only benchmark on single nucleotide variants. Therefore, I anticipate that the tasks proposed here will be a useful resource for future studies.

**Weaknesses:**

The main weakness of the paper is a lack of comparison to a larger set of baselines, and to tasks that have been studied in previous work, which limit the significance of the work. For example, GPN-MSA (Benegas et al., 2024a) was shown to have some capacity to predict regulatory variant effects, and would be a good model to compare to. In addition, a supervised baseline such as Enformer (Avsec et al., 2021), which was shown in Kao et al. 2024 to outperform HyenaDNA and Nucleotide Transformer for eQTL prediction, would also be a relevant comparison to add. Specifically, showing the performance of these two additional baseline models on the tasks/metrics already evaluated in the paper would be informative. In addition, while the focus on indels is novel and interesting, since the model can also be used to score single-nucleotide variants, it would add value to show performance on SNV prediction tasks, such as the SNV eQTL classification task from the Enformer paper (Avsec et al., 2021).

**Questions:**

- What would the effect be of a more strict PIP cutoff (such as 0.1 or 0.05) for the non-causal set in the eQTL task?
- The description of the TFBS disruption task is a bit unclear as written - are the TFBS knockouts from experimental data, or are the TFBS sequences simply altered in silico? More specific details about the type of data used in this evaluation (i.e. positive and negative sequences) would help clarify the task.

Minor:
- since the causal eQTLs referred to in the abstract and in section 4.2 are determined via finemapping rather than experimentally validated, they should be referred to as putative causal eQTLs.

---

> ### Author Response · Authors · 2024-11-19
>
> We thank Reviewer 2Zmu for their thorough and constructive feedback. We provide detailed responses to the various questions and points they raised.
>
> **R_2Zmu.1: Enformer**
>
> **Enformer**  While sequence-to-activity models like Enformer represent a valuable approach, their applicability differs fundamentally from our evolutionary model in several key aspects:
> - Our model aims to capture evolutionary constraint and sequence variation patterns across mammals without requiring functional genomics data
> - Recent studies by Sasse et al. [2] and Huang et al. [3] demonstrate that models like Enformer have limited performance in explaining expression variation between individuals due to cis-regulatory variants
> - We focus specifically on indel effects across species, while Enformer was primarily designed and validated for SNVs in human sequences
>
> Nonetheless, to alleviate any outstanding concern of the reviewer on that point, we are working on including Enformer on the eQTL benchmark during this rebuttal. Given the aforementioned limitations of Enformer to score indels, we will be using the reported POS of the variant as the variant effect prediction.
>
> **R_2Zmu.2: SNP extension**
>
> While indels and larger variants in noncoding regions, particularly promoters, represent a critical unmet need in variant effect prediction, we acknowledge the reviewer's request and will include SNP analyses in the supplement. This work is underway and should be completed by the end of discussion period. However, we would like to reaffirm that the paper's core contributions lie in addressing the understudied challenge of indel effects, which remains our primary focus.
>
> **R_2Zmu.3: GPN-MSA**
> GPN-MSA, while powerful for SNVs, is not well-suited for scoring indels due to fundamental limitations. Its architecture processes whole-genome alignments with independently-treated positions, making it difficult to handle sequence shifts caused by indels. Additionally, its training objective of predicting individual nucleotides from aligned context does not naturally extend to evaluating sequence length changes.
>
> **R_2Zmu.4: Updates to Causal eQTL analysis**
>
> Thank you for the feedback! We ran additional analyses for the PIP values you suggested in Table A3 and Main Table 1. While the overall conclusions for this analysis remain unchanged, we will include these ablations in appendix to confirm the robustness of our conclusions to these parameters
>
> **R_2Zmu.5: Clarifications regarding TBFS and eQTL benchmarks**
>
> We address both points in our manuscript:
>
> **TFBS Disruption Task**
> - We updated the “Task” description to be more clear in the TFBS Benchmark section of the paper.
>
> **Putative Causal eQTLs**
> - We agree the term "putative" would add clarity and will update accordingly in the abstract and in the description of the
>
> References:
>
> [1] A. Sasse, B. Ng, A. E. Spiro, S. Tasaki, D. A. Bennett, C. Gaiteri, P. L. De Jager, M. Chikina, S. Mostafavi, Benchmarking of deep neural networks for predicting personal gene expression from DNA sequence highlights shortcomings. Nat. Genet. 55, 2060–2064 (2023).
>
> [2] C. Huang, R. W. Shuai, P. Baokar, R. Chung, R. Rastogi, P. Kathail, N. M. Ioannidis, Personal transcriptome variation is poorly explained by current genomic deep learning models. Nat. Genet. 55, 2056–2059 (2023).

---

> > ### Author Response · Authors · 2024-11-23
> >
> > As promised we have some additional results to share. We are excited for further engagements for the remainder of the discussion period.
> >
> > **R_2Zmu.1: Enformer**
> >
> > We prioritized evaluating Enformer on the eQTL task since it was trained on genome-wide expression data (CAGE) from FANTOM5, making it theoretically well-suited for predicting expression-modifying variants. Although Enformer is trained on expression information, we actually saw no improvement from Enformer over LOL-EVE on this benchmark.
> >
> > Methodology
> >
> > We evaluate positional constraint and regulatory complexity using two entropy-based metrics:
> > - Single position entropy: Calculated across all possible nucleotide variants
> > - Multi-position entropy for indels: Averaged across affected positions
> >
> > Results
> >
> > For both entropy-based metrics, we find that Enformer underperforms LOL-EVE on the eQTL benchmark (updated Table A3), highlighting Enformer’s challenges dealing with indels.
> >
> >
> > **R_2Zmu.2: SNP extension**
> >
> > We have conducted an extensive analysis on 41K eQTLs from the same source as indel experiments(GTEX finemapped eQTLS).
> >
> > Results Table A6 demonstrate that PhyloP outperforms other methods in this category and that none of the other gLM do particularly well at scoring SNP eQTLS.

---

> ### Comment · Reviewer_2Zmu · 2024-11-24
>
> Thanks to the authors for their detailed response and updates to the manuscript.
>
> While I find the “Task” description of the TFBS Benchmark section to be greatly improved in clarity, and appreciate the modification to refer to “causal eQTLs” as “putatively causal eQTLs,” I have some concerns about the updated analyses presented:
>
> 1. Updates to Causal eQTL analysis. The authors wrote in their response that “We ran additional analyses for the PIP values you suggested in Table A3 and Main Table 1.” I don’t see any analysis with different PIP cutoffs in those tables, but do see Figure A9 that has an ablation of PIP thresholds, perhaps Figure A9 is what the authors were referring to? Unless you changed the PIP cutoff for the negative set, but did not reflect this change in the methods section? In any case, the analysis presented in Figure A9 is different from what I was suggesting, which was to create a more strict negative set that is more likely to be enriched for non-causal eQTLs by using different PIP thresholds for the positive and negative sets.
> 2. Enformer evaluation. I appreciate the addition of Enformer to the set of models benchmarked against, but find the entropy-based scoring metrics to be non-standard. While I can appreciate that the authors may have chosen these entropy-based metrics in order to compute scores from Enformer in a similar manner to the way scores are being computed from other models, I think that Enformer scores should also be reported using similar metrics to the ones used by the Enformer authors in their paper (Avsec et al., 2021). In addition, Enformer is not currently included in the SNV expansion experiment in section A4.5, but I think it would be valuable to include Enformer as a baseline in this comparison as well. Showing that in your SNV evaluation, Enformer achieves similar performance as what is reported in their original paper (AUC ~0.7), would alleviate concerns that the reported metrics for Enformer are due to non-standard scoring methods.
>
> 	Here is the description of the scoring method from Avsec et al., 2021:
> 	“We predicted the effect of a genetic variant on various annotations by computing a forward pass through the model using the reference and alternative alleles, subtracting their difference, and summing outputs across the sequence to obtain a signed score for each training dataset. We averaged scores computed using the forward and reverse complement sequence and small sequence shifts to the left and right. […] To determine how informative different variant annotations are, we trained separate random forest classifiers for each tissue to distinguish causal from non-causal variants using eight-fold cross-validation.”
>
> 3. CADD baseline. As the authors have added CADD as a baseline in Figure A4 based on other reviewer’s suggestions, and it seems to be one of the better performing methods, it seems a bit misleading to omit it from Figure 2 and not include CADD as a baseline in the other benchmarks as well.
>
> I have updated my score to reflect concerns about the updated Enformer analysis.

---

### Official Review · Reviewer_8yAh · 2024-11-04

**Soundness:** 2
**Presentation:** 3
**Contribution:** 3
**Rating:** 5
**Confidence:** 5

**Summary:**

The authors present a mammalian promoter autoregressive DNA language model for insertion or deletion (indel) variant effect prediction (VEP). The proposed approach is novel in terms of data curation as well as the choice of control tags to condition the model on species and gene information. This paper introduces new relevant benchmarks to assess promoter indel VEP capabilities, where the proposed model outperforms the set of baselines evaluated.

**Strengths:**

- The chosen problem, indel VEP, is an impactful and understudied problem in human genetics. The authors do a very good job motivating this problem and the proposed autoregressive language model.
- The proposed approach is substantially different from current approaches, particularly in data curation and conditioning features, which the authors do a good job motivating. It is refreshing to see new ideas tailored to the biological setting that are not just trying out the latest architecture from natural language processing.
- The authors present interesting new benchmarks covering the spectrum of rare and common variants.
- The model achieves a relatively good performance compared to the chosen baselines.

**Weaknesses:**

- Several key baselines are lacking that are crucial to contextualize the performance of the model. First, I expect a comparison to CADD v1.7 (which should also be described in the background section), an established genome-wide deleteriousness predictor that can handle indels. The CADD website includes a web server to score variants. Second, I expect a comparison to at least one current-generation sequence-to-activity model (e.g. Enformer, Borzoi, Sei). Sei might be easier since it also has a web server to score variants, but if the authors want to keep their SOTA claim they should include Enformer or Borzoi. Note 1: not beating these models would not hurt my recommendation for acceptance, as I believe the ideas in the paper would still be valuable. Note 2: the suggested change to the frequency-based task below would reduce the sample size and facilitate running additional models.
- The authors do a good job motivating the conditioning features (species, clade, gene) but do not present any empirical result, which I expect for recommending acceptance. A fast ablation experiment would be to exclude these features (similar to how “control tag dropout” is done during training) at inference time when running VEP, one at a time, to assess their contribution.
- The authors claim “The superior performance of LOL-EVE over other models can be attributed to several factors that address limitations in existing approaches.”. Given the lack of ablation studies, I would be more cautious (e.g. “may be attributed”) and highlight that further studies are necessary to understand the contributions of the different components such as tokenization, region selection, species selection, etc.
- The TFBS disruption is relevant and interesting but the design choices seem rather arbitrary and unnecessarily complicated. A more natural task would be to distinguish variants that affect the binding of any TF vs. those that don’t affect any TFBS, across all human promoters (JASPAR has such information pre-computed: https://jaspar.elixir.no/genome-tracks/). Even if the authors still want to include the current version, I think this simpler formulation of the task should be included regardless. Moving on to the current formulation, I agree that promoters of low-variance genes will be in expectation more constrained than high-variance genes, but that is also true of highly expressed vs. low expressed genes, or highly conserved CDS vs. not (according to multi-species alignment), or highly constrained CDS vs. not (based on population data, see e.g. GeneBayes). Also, in this formulation the metric should be further clarified, e.g. what exactly is the random baseline? This sentence in particular was hard to understand: “A group counts as correctly predicted when the score is lower. ”. What does it mean that a group is correctly predicted? Wouldn’t this statement apply to the TF rather than the group? The score is lower in what compared to what?
- For frequency-based analysis, the authors compare common (MAF > 5%) vs. low-frequency (MAF < 5%) variants. It is common in the field to contrast common vs. singletons, which is expected to yield stronger signals. See e.g. Borzoi, SpliceAI, GPN, GPN-MSA. This should be reported since it would not require any extra computation, just subsetting the current set. If there’s any limitation with indel calling for singletons, it should be mentioned. An alternative set could be common (MAF > 5%) vs. rare variants (MAF < 0.1%).
- For eQTL analysis, the authors compare PIP > 0.95 vs. PIP < 0.95. It is common in the field to compare PIP > 0.9 vs PIP < 0.01 (see e.g. Enformer, Borzoi). Doing this analysis should not require any extra computation since it’s just subsetting the current sets. Also, the actual, unnormalized AUPRC should also be reported in Table 1 (there’s plenty of space). I suspect this would reveal that none of the models are doing well in this task (which is admittedly difficult) in absolute terms. To this point, the phrase “DNABERT-2 surprisingly shows a negative effect size, suggesting potential limitations in its ability to prioritize causal indel variants in this context.” can be misleading. None of the models seem to be doing well on this task, which again, is very difficult.
- It could be worth discussing the fact that some clades are over-represented in the Zoonomia alignment, and its implications.

Minor comments:
- Ref. to Fig. A2 is incorrectly formatted.
- Typo “Causal varant prioritization” in p. 2
-  ”promoter annotations” p. 5, quote incorrectly formatted.

**Questions:**

Please see the suggestions above.  Additional experiments which would make the paper stronger, but might take too much time:
- Do you recommend your model only for indels or also SNVs? Additional experiments on SNVs would be insightful.
- An ablation study where you train the model on entire genomes (or random regions) instead of promoters would be insightful, although expensive to run.

---

> ### Author Response · Authors · 2024-11-19
>
> We thank Reviewer 8yAh for their thorough and constructive feedback. We provide detailed responses to the various questions and points they raised.
>
> **R_8yAh.1: Comparison with sequence-to-activity models and additional baselines**
>
> As requested CADD has been added to the background section of the paper under “Sequence-to-activity models & Meta Predictors”
>
> **CADD and prediction circularity** We included CADD in our frequency-based indel prioritization task but observed limitations consistent with Grimm et al.'s [1] findings about meta predictor circularity:
>
> - Our frequency-based results (Figure A4) show CADD performs poorly compared to evolutionary approaches
> - As a meta predictor integrating many features(https://cadd.gs.washington.edu/static/ReleaseNotes_CADD_v1.7.pdf), CADD likely suffers from both type 1 and type 2 circularity as described by Grimm et al.
> - Moreover, we tried to get CADD scores for our TFBS evaluation but only received 6% of the data back from their server. This suggests a potential limit in coverage for the model.
>
> **Enformer**  While sequence-to-activity models like Enformer represent a valuable approach, their applicability differs fundamentally from our evolutionary model in several key aspects:
> - Our model aims to capture evolutionary constraint and sequence variation patterns across mammals without requiring functional genomics data
> - Recent studies by Sasse et al. [2] and Huang et al. [3] demonstrate that models like Enformer have limited performance in explaining expression variation between individuals due to cis-regulatory variants
> - We focus specifically on indel effects across species, while Enformer was primarily designed and validated for SNVs in human sequences
>
> Nonetheless, to alleviate any outstanding concern of the reviewer on that point, we are working on including Enformer on the eQTL benchmark during this rebuttal. Given the aforementioned limitations of Enformer to score indels, we will be using the reported POS of the variant as the variant effect prediction.
>
> **R_8yAh.2: LOL-EVE Ablation study**
> - We are currently working on these ablations and will provide them by the end of the week.
>
> **R_8yAh.3: TFBS study**
> 1. Thank you for this thoughtful suggestion. While using JASPAR's pre-computed tracks seems appealing at first glance, there are significant technical challenges in reliably distinguishing true TFBS from background noise in promoter regions, particularly within 1kb of the TSS. Position Weight Matrix (PWM) scanning typically identifies numerous potential binding sites, many of which may be biologically inactive. Recent work by Zabet & Adryan [4] and Deplancke et al. [5] has confirmed that PWM-based predictions alone have high false positive rates, especially in promoter regions which are enriched for sequence motifs.
>
> 2. Clarification of Current Metrics:
> - Random baseline: Since there are 2 groups (consistent/variable expression), random accuracy would be 0.5
> - "A group counts as correctly predicted when the score is lower": This is unclear phrasing. It should specify that for each TF, we expect variants in consistently expressed genes to receive lower (more deleterious) scores than variants in variably expressed genes
> - The prediction is evaluated per TF, not per group. A correct prediction occurs when variants in consistently expressed genes receive lower scores than those in variably expressed genes for that TF.
>
> **R_8yAh.4: Singleton vs Common Variant Experiment**
> This experiment has been added to the text and can be seen in Figure A4. We observe that LOL-EVE still performs the best given these cutoffs.
>
> **R_8yAh.4: Updates to Causal eQTL analysis**
> The PIP cutoff has been updated to PIP > 0.9 vs PIP < 0.01 as well as including the non normalized AUPRC. The results for this are shown in Table A3 and Main Table 1. We observe LOL-EVE results outperform other models.
>
> **R_8yAh.5: SNP extension**
> While indels and larger variants in noncoding regions, particularly promoters, represent a critical unmet need in variant effect prediction, we acknowledge the reviewer's request and will include SNP analyses in the supplement. This work is underway and should be completed by the end of discussion period. However, we would like to reaffirm that the paper's core contributions lie in addressing the understudied challenge of indel effects, which remains our primary focus.

---

> > ### Author Response · Authors · 2024-11-19
> >
> > References:
> >
> > [1]D. G. Grimm, C.-A. Azencott, F. Aicheler, U. Gieraths, D. G. MacArthur, K. E. Samocha, D. N. Cooper, P. D. Stenson, M. J. Daly, J. W. Smoller, L. E. Duncan, K. M. Borgwardt, The evaluation of tools used to predict the impact of missense variants is hindered by two types of circularity. Hum. Mutat. 36, 513–523 (2015).
> >
> > [2] A. Sasse, B. Ng, A. E. Spiro, S. Tasaki, D. A. Bennett, C. Gaiteri, P. L. De Jager, M. Chikina, S. Mostafavi, Benchmarking of deep neural networks for predicting personal gene expression from DNA sequence highlights shortcomings. Nat. Genet. 55, 2060–2064 (2023).
> >
> > [3] C. Huang, R. W. Shuai, P. Baokar, R. Chung, R. Rastogi, P. Kathail, N. M. Ioannidis, Personal transcriptome variation is poorly explained by current genomic deep learning models. Nat. Genet. 55, 2056–2059 (2023).
> >
> > [4] Zabet, N. R., & Adryan, B. (2015). Estimating binding properties of transcription factors from genome-wide binding profiles. Nucleic acids research, 43(1), 84-94.
> >
> > [5] Deplancke, B., Alpern, D., & Gardeux, V. (2016). The genetics of transcription factor DNA binding variation. Cell, 166(3), 538-554.

---

> > > ### Author Response · Authors · 2024-11-23
> > >
> > > As promised we have some additional results to share. We are excited for further engagements for the remainder of the discussion period.
> > >
> > > **R_8yAh.1**: Comparison with sequence-to-activity models and additional baselines
> > >
> > > We prioritized evaluating Enformer on the eQTL task since it was trained on genome-wide expression data (CAGE) from FANTOM5, making it theoretically well-suited for predicting expression-modifying variants. Although Enformer is trained on expression information, we actually saw no improvement from Enformer over LOL-EVE on this benchmark.
> > >
> > > Methodology
> > >
> > > We evaluate positional constraint and regulatory complexity using two entropy-based metrics:
> > > - Single position entropy: Calculated across all possible nucleotide variants
> > > - Multi-position entropy for indels: Averaged across affected positions
> > >
> > > Results
> > >
> > > For both entropy-based metrics, we find that Enformer underperforms LOL-EVE on the eQTL benchmark (updated Table A3), highlighting Enformer’s challenges dealing with indels.
> > >
> > > **R_8yAh.2**: LOL-EVE Ablation study
> > >
> > > The LOL-EVE ablation experiments requested will be delivered by the end of the weekend due to computational constraints. We apologize for this delay.
> > >
> > > **R_8yAh.5**: SNP extension
> > >
> > > We have conducted an extensive analysis on 41K eQTLs from the same source as indel experiments(GTEX finemapped eQTLS).
> > >
> > > Results Table A6 demonstrate that PhyloP outperforms other methods in this category and that none of the other gLM do particularly well at scoring SNP eQTLS.

---

> > > > ### Comment · Reviewer_8yAh · 2024-11-25
> > > >
> > > > Thank you for considering my suggestions. I would first like to point out that right now the manuscript does not seem to be in a consistent state.  For example:
> > > >
> > > > > R_8yAh.4: Updates to Causal eQTL analysis The PIP cutoff has been updated to PIP > 0.9 vs PIP < 0.01 as well as including the non normalized AUPRC. The results for this are shown in Table A3 and Main Table 1.
> > > >
> > > > However, Section 4.2 still still talks about a PIP threshold of 0.95.
> > > >
> > > > > R_8yAh.4: Singleton vs Common Variant Experiment This experiment has been added to the text and can be seen in Figure A4
> > > >
> > > > Still, I don’t see any singleton comparison in Figure A4.
> > > >
> > > > Therefore, I will not attempt to give a full revised review at this point. I would still like to point out some major comments:
> > > > 1. I strongly agree with Reviewer 2Zmu in how Enformer variant scores should be computed (calculating delta in activity rather than any entropy measure). Such predictions are even pre-computed for common variants, including indels (link in their code availability section).
> > > > 2. It would be insightful to include GPN-MSA (precomputed scores available) on the SNV evaluation. It would also be helpful to include an evaluation of rare vs. common SNV prioritization.
> > > > 3. The TFBS disruption task is original, but, as the first of its kind, it especially requires more analysis. How do the results look if you contrast the top 100 vs. bottom 100 genes, or top 1000 vs. bottom 1000 genes?

---

### Official Review · Reviewer_mj1m · 2024-11-07

**Soundness:** 2
**Presentation:** 3
**Contribution:** 3
**Rating:** 3
**Confidence:** 5

**Summary:**

This paper presents a generative model that facilitates to score
presence vs absence of indel variants in particular. The generative
model is autoregressive in nature (derived from CTRL). The
probability to generate the next character in a promoter sequence
depends on the already generated subsequence, the clade and the
species to which the sequence is assigned, as well as the gene that
the promoter supports (gene coded as ESM2 embedding of the
corresponding protein sequence). To the best of my understanding,
the achievement of this paper is to provide a (transformer) decoder-only
type approach to computation of probabilities on gaps / insertions
appearing in viable DNA.

**Strengths:**

The paper appears to present a professionally executed study. Choice of data and techniques involved in the approach are of high quality, reflecting recent approved methods. The novelty consists in presenting a transformer decoder-only type of approach by which to compute likelihoods for gaps appearing in DNA. The presentation of the contents has improved considerably, and the paper is now easy to follow and understand, also supported by lovely illustrations.

**Weaknesses:**

There are fairness issues with respect to selection of benchmarking methods and the way they are used. The authors evaluate other methods in a way that is neither recommended by the authors of the respective papers, nor considered in subsequent literature contributions (relating to HyenaDNA and Caduceus). The authors also miss to evaluate the likely strongest competitor (GPN).

**Questions:**

After ample exchange with the authors, there are no questions remaining. The authors responded to my input, although very critical, always in a fair and forthcoming manner, for which I would like to thank.

**Details Of Ethics Concerns:**

Ethics concerns refer to the not yet mature way the authors selected benchmarking methods and evaluated the chosen ones.

---

> ### Author Response · Authors · 2024-11-19
>
> We thank the reviewer for their thoughtful feedback. We address all points of feedback below, including significantly enhanced figures and text edits to address clarification requests. Please let us know if we can help clarify any other aspect of our work during the discussion period.
>
> **R_Mj1m.1: Updates to Model Architecture Figure**
>
> Workflow & Pre-training/Zero-shot Prediction: Thank you for the great suggestions. We fully revamped our Figure 1 accordingly. It now more clearly illustrates our three-stage approach as follows:
>
> 1. Data preprocessing:
> - We extract 1kb promoter sequences upstream of first exons across mammals, grouping species into clades and adding control codes.
>
> 2. Pre-training:
> - LOL-EVE performs next-token prediction conditioned on:
>   - Previous sequence context (x<i)
>   - Clade (c) , species (s) , and ESM gene embeddings (g) control codes
>
> 3. Zero-shot inference
> - For each benchmark task, we:
>   - Generate reference and variant sequences
>   - Calculate probability scores using equation (3)
>   - Compare scores to identify variants with stronger effects
>
> **R_Mj1m.2: Variant Effect Score Clarification**
>
> The score in equation (3) represents the log-likelihood ratio between variant and wildtype sequences. This captures how likely/unlikely the variant sequence is compared to wildtype, given evolutionary patterns learned during pre-training. For rare variants (4.1), lower scores indicate sequences that deviate more from evolutionarily favored patterns, which aligns with rare variants being under stronger negative selection. We edited the text directly after the introduction of equation 3 to clarify this point as suggested.
>
>
> **R_Mj1m.3: Fair Model Comparison**
>
> - We ensure fair comparison by:
>   - Using consistent scoring methods across models (detailed in A.3)
>   - Applying identical preprocessing and evaluation pipelines
>   - Focusing on zero-shot performance without task-specific training
> - We edited the text in the first paragraph of Section 4 to clarify this point as suggested.

---

> > ### Comment · Reviewer_mj1m · 2024-11-22
> >
> > I would like to thank the authors for their answer, and for the improvements in terms of illustrations provided in particular. I am now considerably more confident to understand the manuscript also in its details. As for 'variant effect scores', I now understand that "effect" follows the statistical meaning, I find this confusing, because "variant effect" had me think that your score was referring to phenotypic consequences, so reflecting the layman's meaning of "effect".
> >
> > Unfortunately, my worries about the fairness in experimenting with other methods now, that I understand what is going on, have been reinforced.
> >
> > For example:
> >
> > * HyenaDNA (see 3.3 in https://arxiv.org/pdf/2306.15794) use "soft prompting" as a technique that does justice to dealing with downstream tasks.
> > * Caduceus has clear recommendations for downstream usage of their models; see towards the end of 4.1 and 4.2, see in particular 5.2.3
> >
> > Never in the respective publications, the authors recommend to use their models the way you do it.
> >
> > You do not use the models as intended by the authors, but you distort the meaning of intermediate output to establish "fairness" in terms of the other models following the workflow you yourself *have* to follow. Have to because you do not offer downstream techniques that seamlessly add to the pre-training phase, as HyenaDNA and Caduceus can offer.
> >
> > The bottom line is: your comparisons are very obviously unfair: you use the earlier suggested methods in mistaken ways. This way, you have them look like clearly inferior.
> >
> > Note as well, that your decoder-only architecture effectively prevents what the encoder-only architectures have to offer: they can embed sequences via pre-training first, and construct classifiers based on the embeddings obtained.
> >
> > The one advantage you have is to be able to produce meaningful probabilities, as per the general strategies that decoder-type architectures offer. My recommendation would be to design experiments that encoder-type architectures have trouble dealing with, instead of turning encoder-type architectures into pseudo decoder-type models the way you do it.
> >
> > I consider it therefore essential that you use the methods as the respective suggest to use them.

---

> > > ### Author Response · Authors · 2024-11-23
> > >
> > > We appreciate the reviewer's continued engagement during the discussion period.
> > >
> > > Regarding the baseline implementations for Caduceus and HyenaDNA, our evaluation methodology aligns with established practices in zero-shot variant effect prediction using encoder architectures. In particular, the use of pseudo-perplexities and other masked language modeling heuristics have become standard practice for variant effect prediction, since being introduced for protein language models [1-3] and subsequently adopted for DNA language models [4-7]. Our view is that these two models are expected to be included as baselines in our work, and our implementation follows established practices from numerous prior publications.
> > >
> > > While these two baselines add to our evaluation, the core contributions and claims of our work are independently validated through comprehensive comparisons against more than 10 other baseline methods. We specifically invite the AC and other reviewers to weigh in on this methodological choice: if there is agreement that including Caduceus and HyenaDNA with our current evaluation approach raises concerns, we will remove these comparisons from the manuscript. We would greatly appreciate explicit guidance on this point to ensure our final evaluation aligns with the community's expectations.
> > >
> > > As we believe we have addressed your original concerns otherwise, we would appreciate your feedback on whether there are any remaining substantive issues that prevent you from revising your assessment of our work.
> > >
> > > [1] J. Meier, R. Rao, R. Verkuil, J. Liu, T. Sercu, A. Rives, “Language models enable zero-shot prediction of the effects of mutations on protein function” in Advances in Neural Information Processing Systems, M. Ranzato, A. Beygelzimer, Y. Dauphin, P. S. Liang, J. W. Vaughan, Eds. (Curran Associates, Inc., 2021)vol. 34, pp. 29287–29303.
> > >
> > > [2] N. Brandes, G. Goldman, C.H. Wang et al. Genome-wide prediction of disease variant effects with a deep protein language model. Nat Genet 55, 1512–1522 (2023). https://doi.org/10.1038/s41588-023-01465-0
> > >
> > > [3] J. Cheng, G. Novati, J. Pan, C. Bycroft, A. Žemgulytė, T. Applebaum, A. Pritzel, L. H. Wong, M. Zielinski, T. Sargeant, R. G. Schneider, A. W. Senior, J. Jumper, D. Hassabis, P. Kohli, Ž. Avsec, Accurate proteome-wide missense variant effect prediction with AlphaMissense. Science 381 (2023).
> > >
> > > [4] G. Benegas, S.S. Batra, Y.S. Song, DNA language models are powerful predictors of genome-wide variant effects, Proc. Natl. Acad. Sci. U.S.A. 120 (44) e2311219120, https://doi.org/10.1073/pnas.2311219120 (2023).
> > >
> > > [5] Frederikke Isa Marin, Felix Teufel, Marc Horlacher, Dennis Madsen, Dennis Pultz, Ole Winther, and Wouter Boomsma. BEND: Benchmarking DNA language models on biologically meaningful tasks. In The Twelfth International Conference on Learning Representations, 2024. URL https://openreview.net/forum?id=uKB4cFNQFg.
> > >
> > > [6] Pooja Kathail, Ayesha Bajwa, Nilah M. Ioannidis Leveraging genomic deep learning models for non-coding variant effect prediction. https://arxiv.org/abs/2411.11158
> > >
> > > [7]G. Benegas, C. Ye, C. Albors, J. C. Li, Y. S. Song, Genomic Language Models: Opportunities and challenges, arXiv [q-bio.GN] (2024). http://arxiv.org/abs/2407.11435.

---

> > > > ### Comment · Reviewer_mj1m · 2024-11-23
> > > >
> > > > Thanks for sharing all the publications. I had a more thorough look at all of them in the last couple of hours.
> > > >
> > > > [1] evaluate protein language models with or without downstream adaptation. They conclude that on average (sometimes substantial) improvements can be observed when finetuning / training classifiers was done in addition to pre-training. It is surprising to see that zero-shot prediction does perform fairly well, which appears to be the take-home message of the paper. Importantly, [1] is from 2021, so does not evaluate SSM type models (like Hyena, Mamba/Caduceus) using zero-shot prediction.
> > > >
> > > > [2] use ESM (transformer based) in zero-shot prediction in the frame of a large-scale study
> > > >
> > > > [3] AlphaMissense does invest in downstream adaptation, so is not zero-shot prediction based
> > > >
> > > > [4] present GPN, as a zero-shot predictor based on a transformer based architecture. The novelty is 'dilated convolution'. GPN comes, to the best of my understanding, closest to what you should compare with: the authors suggest to use it this way. In particulary, GPN outperforms PhyloP, which you evaluate as well, and which comes closest to your performance.
> > > >
> > > > [5] is the first citation, in which SSM's (there: HyenaDNA) are looked at. They write (see 2.3.1 in that paper): "HyenaDNA models were evaluated by fine-tuning on NT’s supervised tasks and the Genomic Benchmarks (Greˇ sov´ a et al., 2023) collection" They never run HyenaDNA as a zero-shot predictor, instead they evaluate it as suggested by the authors.
> > > >
> > > > [6] is a review that runs no experiments. Nowhere, they suggest to use SSM type approaches as zero-shot predictors.
> > > >
> > > > [7] is another review, from the authors of GPN. Also this review does not suggest to use SSM's the way you do it.
> > > >
> > > > In SUMMARY: I still find the selection of benchmarked tools both in selecting them (in line with some of the other reviewers, for what I remember) and how you run them, questionable. It looks like an immature and potentially flawed (because indeed unfair) selection. Quite obviously, for example, the pseudo-likelihoods SSM type models compute are not apt for being used immediately in zero-shot prediction frames (at least there is no evidence in the literature whatsoever). Please find a way to run GPN. Please evaluate further transformer based architectures that have been used the zero-shot prediction way. The manuscript requires a major revision with respect to that.
> > > >
> > > > Also, you are writing above that you compare with "10 other baseline methods". In Tables and Figures of the manuscript, I am only counting 5: PhyloP, NT, DNABERT2, HyenaDNA and Caduceus. What other methods did you consider (and where)?
> > > >
> > > > A conclusive positive remark: I clearly see the value of introducing a decoder based model for zero-shot classification in general, and I keep appreciating your contribution with respect to this.

---

> > > > > ### Author Response · Authors · 2024-11-24
> > > > >
> > > > > Thank you for your continued engagement.
> > > > >
> > > > > There appears to be a fundamental misunderstanding. Your initial concern questioned the soundness of using pseudo-perplexity heuristics to evaluate the performance of encoder architecture (trained with MLM objectives) on zero-shot mutation effects tasks. Our response provided examples from literature demonstrating this as an established practice, supporting the validity of our evaluation approach. We should also note that there seems to be a mischaracterization of AlphaMissense as it is specifically focused on zero-shot mutation effects prediction.
> > > > >
> > > > > After careful review of your response, it appears the remaining concern centers on our evaluation of two SSM models (Caduceus and HyenaDNA):
> > > > >
> > > > >    1. For Caduceus, which uses an SSM-based architecture trained with the MLM objective, we maintain that the validity of pseudo-perplexity heuristics is determined by the training objective rather than the architecture. Whether a model uses a transformer, CNN, or SSM architecture is less relevant than its training objective – and as noted in our earlier response, there is substantial precedent for using perplexity in zero-shot evaluations of MLM-trained models. We also invite the reviewer to read ProtMamba [1] -- a SSM-based architecture trained on a closely-related FIM objective -- which uses similar heuristics for zero-shot effects predictions.
> > > > >
> > > > >    2. For HyenaDNA, an SSM-based architecture trained with an autoregressive objective, we evaluate using perplexity metrics – a standard approach for zero-shot predictions, regardless of the underlying architecture. For instance, the recently published Evo model [2] has a SSM-based architecture and uses a similar approach to how we evaluate HyenaDNA for zero-shot tasks.
> > > > >
> > > > > We note that Evo was specifically trained on prokaryotic genomes and intentionally excludes eukaryotic data, making it unsuitable for our human promoter-focused tasks. Similarly, while we considered including GPN and GPN-MSA, these would be inappropriate comparisons since GPN was specifically developed for Arabidopsis thaliana (a plant model organism) and GPN-MSA was specifically developed and benchmarked for single nucleotide variant prediction using aligned sequences, making them unsuitable for our indel-focused mammalian promoter benchmarks.
> > > > >
> > > > > We maintain that our evaluations are methodologically sound, have been used in several publications before us in similar ways, and provide valuable comparative insights. Since this appears to be the only remaining critique the reviewer shared about our work, and given that other reviewers have not raised similar concerns, we would welcome their perspectives on this matter. As previously offered, we are open to removing either baseline if there is broader consensus.
> > > > >
> > > > > It is worth noting that removing these two baselines would not affect our paper's claims, as our evaluation includes a comprehensive set of other baselines from different model families and methodological approaches (see for instance Table A3: nucleotide transformer (x4), DNABERT, GC change, distance to TSS, fathm_indels, PhyloP(x3), enformer (x2)). Moreover, both Caduceus and HyenaDNA show middle-range performance across our three benchmarking tasks, making their inclusion or exclusion immaterial to our conclusions.
> > > > >
> > > > > Given your positive assessment of our core contribution in introducing a decoder-based model for zero-shot classification, we would appreciate your perspective on whether this methodological point about two auxiliary baselines substantially impacts your overall evaluation of our work.
> > > > >
> > > > > [1] D. Sgarbossa, C. Malbranke, A.-F. Bitbol, ProtMamba: a homology-aware but alignment-free protein state space model, bioRxiv (2024)p. 2024.05.24.595730.
> > > > >
> > > > > [2] Nguyen, E., Poli, M., Durrant, M.G., Kang, B., Katrekar, D., Li, D.B., Bartie, L.J., Thomas, A.W., King, S.H., Brixi, G. and Sullivan, J., 2024. Sequence modeling and design from molecular to genome scale with Evo. Science, 386(6723), p.eado9336.

---

> > > > > > ### Comment · Reviewer_mj1m · 2024-11-24
> > > > > >
> > > > > > Thanks another time for addressing my comment.
> > > > > >
> > > > > > Misunderstanding: I don't think there is a misunderstanding. I quite simply disagree with your attitude towards what is allowed and what not relative to fairness concerns when selecting tools to benchmark, and how to employ them. I would never dare to evaluate tools in a way that predominantly favors the presentation of my own tool, if this particular way has not been suggested by either the authors or an affirmative follow-up study.
> > > > > >
> > > > > > AlphaMissense:
> > > > > >
> > > > > > I disagree. Let me cite from their section "AlphaMissense: Fine-tuning AlphaFold for variant effect prediction", which already says in the title that they finetune other than you claim: "... AlphaMissense is trained in two stages. ... After pretraining, the masked language modeling head can already be used for variant effect prediction by computing the log-likelihood ratio between the reference and alternative amino acid probabilities ..." So far, so good. But then, immediately thereafter: "In the second stage (Fig. 1A), the model is fine-tuned on human proteins with an additional variant pathogenicity classification objective..."
> > > > > >
> > > > > > In summary, your claim that AlphaMissense is used for zero-shot prediction without finetuning is misleading.
> > > > > >
> > > > > > ProtMamba: As you rightfully say, ProtMamba trains Mamba on the FIM objective (which already is arguably substantially different, because it trains to model to insert known context. Unlike you say, they do invest in a finetuning strategy, see 2.2: "Both models were fine-tuned for 2 days on predicting only the FIM amino acids to improve inpainting capabilities, yielding the models ProtMamba/ProtMamba Long, Fine-tuned." Note further that mostly the finetuned versions of their models provided optimal results.
> > > > > >
> > > > > > GPN: You made it possible to use PhyloP for indels instead of SNP's. I am convinced similar strategies may apply for GPN, so If there are concerns for GPN, likely there are concerns for PhyloP as well. While GPN was trained on Arabidopsis in the own paper, they provide easy-to-use code for training it on your own data, see https://github.com/songlab-cal/gpn docu section "Training on your own data".
> > > > > >
> > > > > > In summary: removing HyenaDNA and Caduceus from your evaluation would eliminate my fairness concerns. Adding GPN (and possibly other tools the other reviewers suggested) would leave me with no doubts that you compared with the state of the art, and not only tools that were shown to underperform in prior work.

---

### Official Review · Reviewer_SwRw · 2024-11-08

**Soundness:** 2
**Presentation:** 3
**Contribution:** 2
**Rating:** 3
**Confidence:** 4

**Summary:**

LOL-EVE is an autoregressive language model trained on mammalian promoter sequences that operates on a single base pair resolution. The authors focus on promoter indels and construct a benchmark containing three tasks to evaluate model performance: (1) distinguishing common and rare indels, (2) prioritizing fine-mapped eQTLs, and (3) distinguishing TFBS mutation scores between low-variability and high-variability genes. The authors compare LOL-EVE to phyloP and other commonly used genomic language models and find that their method outperforms the rest on these three tasks.

**Strengths:**

- Instead of training a genome-wide language model which has been shown to have reduced performance in understanding any one genomic element, the authors train only on proximal promoter sequences, which likely have a well-conserved, cell-type agnostic cis-regulatory grammar. This is the ideal setting to use self-supervised language models.
- The authors use an autoregressive model, with individual base pairs as tokens, which is relatively uncommon in the field but allows them to easily make predictions on indels as opposed to just SNVs.
- The authors extracted homologous promoter sequences from whole-genome alignments and then cleverly validated that aligned sequences in other species were likely promoters by using Sei promoter scores.
- The authors developed a biologically-meaningful benchmark for a setting in which benchmark datasets have not yet been constructed.

**Weaknesses:**

1. While the tasks are well-motivated, they may inadvertently emphasize gene-specific rather than variant-specific properties.
     - Frequency-based indel prioritization task: Models are asked to distinguish rare promoter indels from common ones. However, if a gene is under selective pressure, the number of common promoter indels will decrease. Accordingly, a model might achieve high performance by recognizing gene-level constraints instead of specific properties of each indel variant. To avoid this, metrics should be calculated separately for each gene and then averaged.
     - Causal eQTL prioritization: The negative set consists of non-finemapped eQTLs that may not be matched to the same genes as the positive set. This again allows models to potentially rely on gene-related features rather than focusing on the distinct attributes of each variant. To avoid this, the negative set should be matched to be from the same genes as the positive set.
     - TFBS disruption: Performance is evaluated by comparing indel scores between low-variability and high-variability genes. Low-variability genes are generally under greater evolutionary constraints, which means that models might perform well by understanding these gene constraints instead of learning about the specific effects of the variants within the TFBS. This task seems to be deliberately designed to test understanding of gene constraint, so it doesn't need to be changed.

A model that learns how much variability there is in a promoter across species (which LOL-EVE likely does) might understand selective pressures on the gene and do well on these tasks without really understanding a good causal cis-regulatory grammar.

2. The authors should compare their method to sequence-to-activity models, like Enformer. While the authors claim that functional genomics data is not often available to train sequence-to-activity models, this data is readily available for humans at least. They should also compare to simpler non-genomic language model baselines, like CADD, fathmm-indel, and deltaSVM. Even simple baselines like GC content of the surrounding base pairs or length of the indel might be competitive.

3. The authors should report performance on promoter SNVs even if the primary goal of developing the autoregressive model was to handle indels. Even if LOL-EVE does not outperform existing methods on SNVs, knowing SNV results would help readers better understand the limitations and strengths of this model.

**Questions:**

- phyloP scores were extracted for the single base pair position of the indel, presumably corresponding to the position of the leftmost element of the sequence. However, this doesn't capture the constraints of other base pairs removed in a deletion. How do performance metrics change if all reference positions in a deletion are incorporated? phyloP base pair statistics are also noisy, so it might be worth averaging scores in a 10 or 25 base pair region surrounding the indel.
- A paper using ESM for scoring protein indels (https://www.nature.com/articles/s41588-023-01465-0#Sec8) suggests that length-normalizing pseudolikelihoods is not the best way to score indels in masked language models. How do your results change if you get rid of the length-normalization?
- I don't understand how you identified deletions in TFBS for the TFBS disruption task? Did these deletions need to be occurring in gnomAD? Did the deletions span the entire TFBS?

---

> ### Author Response · Authors · 2024-11-19
>
> We thank Reviewer SwRw for their thorough and constructive feedback. We provide detailed responses to the various questions and points they raised.
>
> **R_SwRw.1: Gene-level confounding in frequency-based indel prioritization in Frequency-based indel prioritization task and Causal eQTL prioritization.**
>
> We appreciate this comment that has the intent to improve our analysis. We agree with the reviewer about the potential risk of models exploiting gene-level selective constraints rather than variant-specific properties. The main challenge with computing metrics at the gene-level is data sparsity - there is not enough data for the majority of genes (no gene has even 100 labels per class), which would result in unreliable estimates, as shown in Figures A5 and A7. However, to address these concerns, we analyzed LOL-EVE score distributions across genes with sufficient variants (≥12) and found substantial variant-level variation within genes, with genes on average capturing 62.6% of the total score range (Figure A6), suggesting our model learns more than just gene-level features.
>
> **R_SwRw.2: Comparison with sequence-to-activity models and additional baselines**
>
> 1. **CADD and prediction circularity.** We included CADD in our frequency-based indel prioritization task but observed limitations consistent with Grimm et al.'s [1] findings about meta predictor circularity:
> - Our frequency-based results (Figure A4) show that CADD performs poorly compared to evolutionary approaches
> - As a meta predictor integrating many features, CADD likely suffers from both type 1 and type 2 circularity as described by Grimm et al.
> - Moreover, we tried to get CADD scores for our TFBS evaluation but only received 6% of the data back from their server. This suggests a potential limit in coverage for the model.
>
> 2. **FATHMM-indel** We included FATHMM-indel in the causal eQTL and TFBS disruption benchmarks, which required porting these analyses to GRCh37 coordinates [Figures A10 and Table A3]. In both cases LOL-EVE significantly outperforms this baseline. Despite coordinating with the tool authors to process variants in small batches, server limitations prevented us from including FATHMM-indel in the frequency benchmark due to the sheer size of the dataset.
>
> 3. **Simple baselines** As suggested by the reviewer, we also included GC content differences between the variant and wild type sequence and distance to TSS as baseline features. Please see Figure A4, Table A3, and A10. Indel length was considered in analysis for Figure A4 (variant prioritization task) and A10 (TBFS task), and included as a cutoff for Table A3 (eQTL task). Using indel cutoffs for smaller indels in the causal eQTL task was problematic due to severe class imbalance (a large majority of background variants are present in the smaller indel length bins). This is illustrated in Figure A8. Consequently, we report aggregated performance for indels with length above 5, and observe similar conclusions as reported in the original manuscript.
>
> 4. **Enformer** While sequence-to-activity models like Enformer represent a valuable approach, their applicability differs fundamentally from our evolutionary model in several key aspects:
> - Our model aims to capture evolutionary constraint and sequence variation patterns across mammals without requiring functional genomics data
> - Recent studies by Sasse et al. [2] and Huang et al. [3] demonstrate that models like Enformer have limited performance in explaining expression variation between individuals due to cis-regulatory variants
> - We focus specifically on indel effects across species, while Enformer was primarily designed and validated for SNVs in human sequences
>
> 	Nonetheless, to alleviate any outstanding concern of the reviewer on that point, we are working on including Enformer on the eQTL benchmark during this rebuttal. Given the aforementioned limitations of Enformer to score indels, we will be using the reported POS of the variant as the variant effect prediction.
>
> **R_SwRw.3: SNP extension**While indels and larger variants in noncoding regions, particularly promoters, represent a critical unmet need in variant effect prediction, we acknowledge the reviewer's request and will include SNP analyses in the supplement. This work is underway and should be completed by the end of discussion period. However, we would like to reaffirm that the paper's core contributions lie in addressing the understudied challenge of indel effects, which remains our primary focus.

---

> > ### Author Response · Authors · 2024-11-19
> >
> > **R_SwRw.4: Improving indel scoring with multi-position conservation metrics**
> >
> > Thank you for this suggestion - we will include these window-based analyses in our revision. However, we note that such approaches require arbitrary choices about window sizes and how to potentially combine multiple signals, further highlighting the fundamental limitations of existing approaches for indels effect prediction. In contrast, our autoregressive approach offers a more principled solution by naturally modeling joint probability distributions over nucleotides, enabling direct scoring of sequence changes without such heuristics.
> >
> > **R_SwRw.5: Length Unnormalized scores**
> >
> > This analysis is underway across all models -- we aim to complete it by the end of the week.
> >
> > **R_SwRw.6: Clarification of TFBS disruption methodology**
> >
> > Thank you for the feedback - we have consequently expanded our methods section to provide a more detailed description of our approach, as follows:
> >
> > 1. TFBS deletion generation process:
> >
> > - We performed in silico deletions rather than using natural variants
> > - For each TFBS identified by JASPAR PSSMs (score > 0.8):
> >   - We created synthetic deletion spanning the entire TFBS
> >   -  We scored the mutated sequence with the same PSSM
> >   -  We considered TFBS disrupted if score drops below 0.8 threshold
> > - This approach allows systematic evaluation of binding site disruption
> >
> > 2. Analysis details:
> > - We analyzed 340 TFs expressed in >30 GTEx tissues
> > - We generated 38,854 deletions in consistently expressed genes
> > - We generated 3,790 deletions in variably expressed genes
> > - Each deletion precisely matches TFBS boundaries identified by PSSM
> > References:
> >
> > [1] D. G. Grimm, C.-A. Azencott, F. Aicheler, U. Gieraths, D. G. MacArthur, K. E. Samocha, D. N. Cooper, P. D. Stenson, M. J. Daly, J. W. Smoller, L. E. Duncan, K. M. Borgwardt, The evaluation of tools used to predict the impact of missense variants is hindered by two types of circularity. Hum. Mutat. 36, 513–523 (2015).
> >
> > [2] A. Sasse, B. Ng, A. E. Spiro, S. Tasaki, D. A. Bennett, C. Gaiteri, P. L. De Jager, M. Chikina, S. Mostafavi, Benchmarking of deep neural networks for predicting personal gene expression from DNA sequence highlights shortcomings. Nat. Genet. 55, 2060–2064 (2023).
> >
> > [3] C. Huang, R. W. Shuai, P. Baokar, R. Chung, R. Rastogi, P. Kathail, N. M. Ioannidis, Personal transcriptome variation is poorly explained by current genomic deep learning models. Nat. Genet. 55, 2056–2059 (2023).

---

> > > ### Author Response · Authors · 2024-11-23
> > >
> > > As promised we have some additional results to share. We are excited for further engagements for the remainder of the discussion period.
> > >
> > > **R_SwRw.2**: Comparison with sequence-to-activity models and additional baselines
> > >
> > > We prioritized evaluating Enformer on the eQTL task since it was trained on genome-wide expression data (CAGE) from FANTOM5, making it theoretically well-suited for predicting expression-modifying variants. Although Enformer is trained on expression information, we actually saw no improvement from Enformer over LOL-EVE on this benchmark.
> > >
> > > Methodology
> > >
> > > We evaluate positional constraint and regulatory complexity using two entropy-based metrics:
> > > - Single position entropy: Calculated across all possible nucleotide variants
> > > - Multi-position entropy for indels: Averaged across affected positions
> > >
> > > Results
> > >
> > > For both entropy-based metrics, we find that Enformer underperforms LOL-EVE on the eQTL benchmark (updated Table A3), highlighting Enformer’s challenges dealing with indels.
> > >
> > > **R_SwRw.3**: SNP extension
> > >
> > > We have conducted an extensive analysis on 41K eQTLs from the same source as indel experiments(GTEX finemapped eQTLS).
> > >
> > > Results Table A6 demonstrate that PhyloP outperforms other methods in this category and that none of the other gLM do particularly well at scoring SNP eQTLS.
> > >
> > > **R_SwRw.4**: Improving indel scoring with multi-position conservation metrics
> > >
> > > As suggested, we have expanded our analysis using PhyloP conservation scores spanning different windows around the variant position. We use the median rather than mean for PhyloP window scores because PhyloP follows a bimodal distribution of conservation (positive) and acceleration (negative) scores. The median provides a more robust measure of central evolutionary constraint that is less sensitive to extreme values and better represents the typical selective pressure across the window.
> > >
> > > Methodology
> > >
> > > - For each variant, we calculated conservation metrics using two window sizes: 10bp and 30bp centered on variant position
> > > - Median PhyloP scores were computed across all valid bases within each window
> > > - Missing PhyloP values were excluded to prevent bias in conservation estimates
> > >
> > > Results
> > >
> > > Our updated analysis shows:
> > > - eQTL task: No improvement over LOL-EVE (Updated Table A3)
> > > - TFBS task: No improvement over LOL-EVE (Updated Figure A10)
> > > - Frequency task: No improvement over LOL-EVE (Updated Figure A4)
> > >
> > > **R_SwRw.5**: Length Unnormalized scores
> > >
> > > Thank you for the excellent suggestion. We indeed observe slightly better aggregate performance when removing length normalization, as similarly observed in the paper referenced by the reviewer. More specifically,LOL-EVE's performance improves when length normalization is removed across the Frequency-based indel prioritization task and TFBS disruption tasks (Figure A11, A12) but does comparably for the Causal eQTL prioritization task as shown in Table A5.

---

> ### Comment · Reviewer_SwRw · 2024-11-25
>
> Thank you for the clarifications and comparisons to additional methods. As other reviewers have noted, there seems to be a disconnect between the new figures added to the appendix and the original figures/framing in the main paper. I assume that this is because of the lack of time and that this will be fixed before the discussion period ends. Assuming those discrepancies are addressed, I still have three primary concerns, ordered by importance.
>
> * Comparison to CADD [high importance]: The 11/19 version of the paper showed that CADD significantly outperformed LOL-EVE on the causal eQTL classification task. This was unexplainably removed in the 11/22 version. I am not sure if this is because of the circularity issue mentioned by the authors. However, after re-reading Grimm et al., it is still unclear to me how CADD suffers from either the type 1 or type 2 circularity issue on these specific tasks (which have nothing to do with missense variant effect prediction). Correct me if I'm wrong but the regulatory predictions that CADD uses as features do not seem to come from models supervised on fine-mapped eQTL data.
>
> * Comparison to Enformer [high importance]: As other reviewers have noted, the method to evaluate Enformer on indels is non-standard and not clearly explained. A very simple metric, if you do not want to use the random forest method used by Enformer and proposed by Reviewer 2Zmu, is to just average the |ISM| scores for all accessibility or CAGE output tracks.
>
> * Gene-level confounding [low importance]: I appreciate the authors' willingness to try to determine if gene-specific metrics could be computed to reduce the impact of confounding by gene-level selective constraints. I disagree, however, with the authors' conclusion that it is infeasible to get rid of this confounding. Based on the number of indels per gene presented in Fig. A5, I do think you could compute frequency-based indel prioritization metrics on the always >50 genes with >= 10 variants. While the metric per gene might be noisy, you could average over genes to get a pretty stable metric. For the causal eQTL prioritization task, it seems like you could subset to putatively causal eQTLs with a negative (background) eQTL in the same gene. Then, having n_{g} positive and n_{g} negative eQTLs for each gene g, you could compute a genome-wide AUROC. Regardless, I don't think this is necessarily of great importance given Fig. A6, especially given the limited time remaining in the discussion.
>
> Overall, I believe that the authors need to include CADD performance metrics and compute Enformer scores in a more standard way before I can recommend acceptance. I encourage the authors--regardless of the results they find--to modify their main text and figures to reflect the performance of the new tools benchmarked and to not obfuscate results that make their model look worse. For now, I will keep my score as is.

---

### Author Response · Authors · 2024-11-19
**Response to all reviewers**

Dear Reviewers,

We sincerely thank you for your insightful and constructive feedback on our manuscript. We have conducted all requested analyses and substantially improved the text as a result. Specifically, we have significantly strengthened our submission along the following aspects:

1. **Additional analyses and ablations**
- We demonstrate our conclusions are robust to the parameters used to define the performance metrics (eg., MAF and PIP thresholds) [Reviewers 8yAh and 2Zmu], and added AUPRC/normalized AUPRC metrics to all benchmarks as suggested  [Reviewers 8yAh].
- To account for gene-level effects, we analyze LOL-EVE score ranges per-gene [Reviewer SwRw].
- While our work has primarily focused on the effects of indels, we are adding the SNV benchmark as suggested. This will be completed and integrated into the manuscript by the end of the week (EOW) [Reviewers SwRw and 2Zmu].
- We will include the unnormalized lol-eve scores to all the benchmarks to confirm the soundness of our length-normalization scheme by EOW [Reviewer SwRw].
- We carried out additional ablations to further substantiate our claims regarding our model design decisions. Final results by EOW [Reviewer 8yAh].

2.**Comparison with new baselines**
- Comparison to CADD [Reviewers SwRw and 8yAh]
- Comparison to Enformer (final results by EOW) [Reviewers SwRw, 8yAh and 2Zmu]
- Comparison to fathmm_indel [Reviewer SwRw]
- Comparison to simple baselines (ie., delta GC content, dist to TSS) [Reviewer SwRw]

3.**Text clarifications and improved figures**
- We edited Figure 1 to improve comprehensiveness and clarity [Reviewer mj1m]
- We clarified the text where suggested, including  the TBFS task description [Reviewers SwRw and 2Zmu], relationships between likelihoods learned by the various models and downstream tasks of interest [Reviewer mj1m]

We highlighted above the few additional analyses that are still in progress, and we will share these additional results upon completion by the end of the week. Please let us know if there is any outstanding point we can help clarify in the meantime.

Kind regards,
The authors

---

### Note · Authors · 2024-12-01

**Comment:**

We thank all reviewers for their time and thoughtful feedback. Unfortunately, we did not have time to complete all experiments requested during the discussion period. We will be incorporating your feedback into a future submission of the paper.

**Withdrawal Confirmation:**

I have read and agree with the venue's withdrawal policy on behalf of myself and my co-authors.